# Group-wise oracle-efficient algorithms for online multi-group learning

**Samuel Deng**
Department of Computer Science
Columbia University
samdeng@cs.columbia.edu

**Daniel Hsu**
Department of Computer Science
Columbia University
djhsu@cs.columbia.edu

**Jingwen Liu**
Department of Computer Science
Columbia University
jingwenliu@cs.columbia.edu

## Abstract

We study the problem of online multi-group learning, a learning model in which an online learner must simultaneously achieve small prediction regret on a large collection of (possibly overlapping) subsequences corresponding to a family of *groups*. Groups are subsets of the context space, and in fairness applications, they may correspond to subpopulations defined by expressive functions of demographic attributes. In contrast to previous work on this learning model, we consider scenarios in which the family of groups is too large to explicitly enumerate, and hence we seek algorithms that only access groups via an optimization oracle. In this paper, we design such oracle-efficient algorithms with sublinear regret under a variety of settings, including: (i) the i.i.d. setting, (ii) the adversarial setting with smoothed context distributions, and (iii) the adversarial transductive setting.

## 1 Introduction

We study the problem of online multi-group learning, originally introduced by [BL20] (adapting the specialists/time-selection setup of [Blu97; Fre+97; BM07]). In this learning model, we consider a collection of groups $\mathcal{G}$, which are (possibly intersecting) subsets of a context space $\mathcal{X}$, as well as a hypothesis class $\mathcal{H}$ of functions defined on $\mathcal{X}$. Contexts $x_1, x_2, \ldots, x_T$ arrive one-by-one over a sequence of $T$ rounds, and the learner must make a prediction associated with each $x_t$. The learner's goal is to perform well on *every* subsequence of rounds corresponding to each group $g \in \mathcal{G}$. Here, performance is measured relative to the predictions of the best-in-hindsight hypothesis $h \in \mathcal{H}$ for the specific subsequence under consideration.

A common interpretation of multi-group learning—which is natural when considering fairness in machine learning (ML)—identifies contexts $x \in \mathcal{X}$ with individuals, each group $g \in \mathcal{G}$ with a subpopulation (perhaps defined by a combination of various demographic features such as age and gender), and each hypothesis $h \in \mathcal{H}$ with a classifier that makes predictions about individuals [RY21]. The goal of the learner, then, is to predict as well as the best subpopulation-specific hypothesis, for all subpopulations simultaneously.

The standard benchmark in online learning is regret, which compares the performance of the learner on all rounds to that of the single best hypothesis in hindsight. But such a benchmark is only meaningful if there is a hypothesis that performs well in all contexts. At the other extreme, we may hope that the learner performs as well as using the best context-specific hypothesis in every

round. However, this may be impossible if no context is ever repeated. The group-wise notion of regret in online multi-group learning naturally interpolates between these two extremes: the former is recovered when $\mathcal{G} = \{\mathcal{X}\}$, and the latter when $\mathcal{G} = \{\{x\} : x \in \mathcal{X}\}$.

We are particularly interested in scenarios where $\mathcal{G}$ may be extremely large (and perhaps even infinite). In such cases, it is too time-consuming to explicitly enumerate the groups in $\mathcal{G}$, and this precludes the use of the algorithmic solutions from prior works [BL20; Ach+23].

The use of highly expressive families of groups has been a recent focus in the ML fairness literature, where fairness with respect to such rich families of groups is seen as compromise between coarse notions of statistical fairness that ignore intersectionality, and individualized notions of fairness that are typically difficult to ensure [Kea+18; Heb+18; KGZ19; GKR22; Glo+23]. For example, if groups are defined by simple combinations of demographic attributes (e.g., linear threshold functions), then the number of subsequences determined by these groups may grow exponentially with the number of attributes. To deal with the intractability of explicit enumeration of groups, these prior works rely on optimization oracles that implicitly search through the family of groups. In this work, we seek to do the same, but for the online (as opposed to batch) problem at hand.

Another motivation for multi-group learning with rich families of groups comes from the literature on "subgroup robustness" [Sag+20], where one is concerned with the test-time distribution shifting from the training distribution by restricting to subsets (or "subgroups") of the feature space. Such scenarios have been practically motivated, for instance, in medical domains where training data sets are constructed to include data from all patients (healthy and sick) as a matter of convenience, but the population relevant to the application is only a subset of the sick patients for which an intervention is potentially possible [Oak+20]. It may be difficult to anticipate which subgroup will ultimately be relevant (or even to provide an explicit shortlist of such subgroups), but multi-group learning with a very rich and expressive family of groups $\mathcal{G}$ ensures that, as long as a subgroup is well-approximated or within the collection $\mathcal{G}$, it obtains our theoretical guarantees. This motivates dealing with large or potentially infinite $\mathcal{G}$ to provide guarantees for as many subgroups as possible.

## 1.1  Summary of Results

We construct an end-to-end oracle-efficient online learning algorithm that is oracle-efficient in *all* the problem parameters. Namely, it is oracle-efficient in both $\mathcal{H}$ and $\mathcal{G}$. In the case of finite $\mathcal{H}$ or $\mathcal{G}$, this admits an exponential computational speedup over the previous algorithms; in the case of infinite $\mathcal{G}$, this is the first algorithm that achieves multi-group online learning. With this in mind, we can even think of $\mathcal{G}$ as representing a binary-valued function class that takes in contexts and selects subsequences $S \subseteq [T]$ based on those contexts in possibly complex ways.

Previous work has shown that the basic goal of designing a computationally efficient algorithm (in all problem parameters) to achieve $o(T)$ regret for fully worst-case adversaries is impossible [HK16; Blo+22]. Research has therefore focused on natural structural assumptions in which computational efficiency and sublinear regret is possible, albeit in the traditional setting without groups [SKS16; Hag+22; HRS22; Blo+22]. Because the multi-group online adversarial setting is strictly harder than the standard setting in which this lower bound applies (simply consider multi-group learning with one group: the entire sequence), we must also take similar assumptions to circumvent the computational hardness result. In this work, we consider the same structural assumptions in the multi-group scenario. Specifically, we make the following contributions (with $\tilde{O}(\cdot)$ suppressing log factors):

1. **Group-wise oracle efficiency for the smoothed setting.** We present an oracle-efficient algorithm that achieves $\tilde{O}(\sqrt{dT/\sigma})$ regret on every group $g \in \mathcal{G}$ for a binary-valued action space for the *smoothed online learning setting* of [HRS22; Blo+22], where $d$ is a bound on the VC dimension of $\mathcal{H}$ and $\mathcal{G}$, and $\sigma$ is a parameter interpolating between the scenarios in which contexts are generated by a worst-case adversary and a benign, fully i.i.d. adversary.
2. **Group-wise oracle-efficiency through generalized follow-the-perturbed leader.** If a sufficient condition referred to as $\gamma$-*approximability* in previous literature ([Wan+22]) is met, a variant of our oracle-efficient algorithm achieves, for each particular $g \in \mathcal{G}$, a regret of $\tilde{O}(\sqrt{NT_g \log |\mathcal{H}||\mathcal{G}|})$ for finite $\mathcal{H}$ and $\mathcal{G}$, where $N$ corresponds roughly to a required number of perturbations, and $T_g$ is the number of rounds $t \in [T]$ in which $x_t \in g$. The dependence on $T_g$ as opposed to $T$ is preferable if some groups do not appear frequently over the $T$ rounds. As a special case, our

| Work | Setting | Regret | Computation | Oracle-efficient in $\mathcal{H}$ | Oracle-efficient in $\mathcal{G}$ |
|---|---|---|---|---|---|
| [BL20] | Adversarial | $\sqrt{T_g \log |\mathcal{H}||\mathcal{G}|}$ for all $g \in \mathcal{G}$ | Time: $O(|\mathcal{H}||\mathcal{G}|)$ Space: $O(|\mathcal{H}||\mathcal{G}|)$ | No | No |
| [Ach+23] | Adversarial | $\sqrt{T_g \log |\mathcal{H}||\mathcal{G}|}$ for all $g \in \mathcal{G}$ | Time: $O(|\mathcal{G}|)$ Space: $O(|\mathcal{G}|)$ | Yes | No |
| [Blo+22] | $\sigma$-Smooth | $\sqrt{\frac{dT \log T}{\sigma}}$ (does not handle multi-group setting) | Time: poly($T$) calls to optimization oracle | Yes | N/A |
| [Hag+22] | $\sigma$-Smooth | $\sqrt{\frac{dT \log T}{\sigma^{1/2}}}$ (does not handle multi-group setting) | Time: poly($T$) calls to optimization oracle | Yes | N/A |
| Ours (Theorem 4.1) | $\sigma$-Smooth | $\sqrt{\frac{dT \log T}{\sigma}}$ for all $g \in \mathcal{G}$ | Time: poly($T$) calls to optimization oracle | Yes | **Yes** |
| Ours (Theorem 5.1) | $\gamma$-approximable | $\sqrt{NT_g \log |\mathcal{H}||\mathcal{G}|}$ for all $g \in \mathcal{G}$ | Time: poly($T$) calls to optimization oracle | Yes | **Yes** |
| Ours (Corollary 5.1.1) | Transductive | $N^{1/4}\sqrt{T_g \log |\mathcal{H}||\mathcal{G}|}$ for all $g \in \mathcal{G}$ | Time: poly($T$) calls to optimization oracle | Yes | **Yes** |

Table 1: In the table above, $T$ is the number of rounds of online learning, $T_g$ is the number of rounds for a particular group $g$, $\mathcal{H}$ is the hypothesis class, $\mathcal{G}$ is the collection of groups, and $\sigma$ is the smoothness parameter defined in Definition 4.1. $d$ is an upper bound on the VC dimension of $\mathcal{H}$. In our $\sigma$-smooth result in the fifth row, $d$ is also an upper bound on the VC dimension of a possibly infinite collection of groups, $\mathcal{G}$. In the $\gamma$-approximable setting of the sixth row, $N$ is the number of perturbations.

algorithm also achieves $\tilde{O}(N^{1/4}\sqrt{T_g \log |\mathcal{H}||\mathcal{G}|})$ in the *transductive setting* of [BKM97; KK05], where the adversary must reveal a set of $N$ future contexts before learning begins.

Table 1 summarizes our results in relation to existing work. Our algorithms follow a more general algorithm design template based on the *adversary moves first (AMF)* framework of [Lee+22]. We extend this framework with the following technical enhancements: sparsifying the implicit distributions over the hypothesis spaces used by *follow-the-perturbed-leader (FTPL)* algorithms, and simplifying the min-max game that is solved in every iteration of AMF.

In particular, the AMF framework allows us to have low-regret with respect to all group-hypothesis pairs; the multi-objective regret guarantees are useful for our purposes because we want to guarantee simultaneous low regret over all $g \in \mathcal{G}$. However, naively applying AMF requires us to enumerate these objectives, in turn enumerating $\mathcal{G}$, the main issue we want to avoid. FTPL comes to the rescue and allows us to have low-regret to all pairs implicitly without enumerating them. So AMF allows us to compete with all $g \in \mathcal{G}$; FTPL ensures this is efficient. The algorithmic details and main technical challenges can be found in Section 4.2. These techniques allow us to adapt this framework to the scenario where the number of groups is too large to enumerate, and they may find use in other multi-objective learning problems.

## 1.2 Related Work

[BL20] showed that it is possible to achieve sublinear multi-group regret by reducing to the *specialists framework* of [Blu97; Fre+97; BM07] (a.k.a. *sleeping experts*). The multi-group regret they achieve scales logarithmically in both $\mathcal{H}$ and $\mathcal{G}$; this recovers the minimax regret when specializing to the standard online learning setting (with finite $\mathcal{H}$) with only a single group. The paper focuses on regret rather than computational considerations; a direct implementation of their algorithm uses time and space linear in $|\mathcal{H}| \times |\mathcal{G}|$, and there are no stated guarantees for infinite $\mathcal{H}$ or $\mathcal{G}$.

[Ach+23] show how to avoid enumeration of $\mathcal{H}$ using an optimization oracle for $\mathcal{H}$. They achieve this by applying a meta-algorithm atop a black-box oracle-efficient online learning algorithm, but this meta-algorithm ultimately requires explicit enumeration of $\mathcal{G}$. Our work, in contrast, uses an optimization oracle for $\mathcal{G} \times \mathcal{H}$ jointly and hence avoids explicit enumeration of either $\mathcal{G}$ or $\mathcal{H}$.

Multi-group (agnostic) learning has also been studied in the batch setting [RY21; TH22; GKR22; HJZ24]. In this setting, training data is drawn i.i.d. from a fixed distribution, and the learner's goal is to find a single hypothesis $\hat{h}$ (possibly outside of $\mathcal{H}$) that ensures small excess risk $\mathbb{E}[\ell(\hat{h}(x), y) \mid x \in g] - \inf_{h \in \mathcal{H}} \mathbb{E}[\ell(\hat{h}(x), y) \mid x \in g]$ for every group $g \in \mathcal{G}$ simultaneously. The works of [RY21; HJZ24] design algorithms for achieving this learning criterion under a certain "multi-PAC compatibility" assumption on $\mathcal{H}$ and $\mathcal{G}$. [GKR22; TH22] design multi-group learning algorithms that remove the need for this assumption. One of the algorithms of [TH22], which enjoys near optimal sample complexity for general but finite $\mathcal{H}$ and $\mathcal{G}$, is based on the the online multi-group learning approach of [BL20] combined with online-to-batch conversion.

The proof of our algorithm builds on two primary technical frameworks studied in previous literature: the *adversary moves first (AMF)* framework of [Lee+22], and a line of work designing follow-the-perturbed leader style algorithms [KV05] for adversarial online learning in the oracle-efficient learning model [KK05; SKS16; Hag+22; Blo+22; Wan+22].

## 2 Preliminaries

### 2.1 Notation

Throughout, $\mathcal{X}$ denotes a *context space*, and $\mathcal{Y}$ denotes an *action space*. For example, in a typical (online) supervised learning setup, $\mathcal{X}$ is the feature space, and $\mathcal{Y}$ is the label space. A *group* $g$ is a subset of the context space $\mathcal{X}$. We overload the notation $g$ for a group by using it as an indicator function $g(x) := \mathbf{1}\{x \in g\}$ for group membership. Let $2^{\mathcal{X}}$ denote all subsets of the context space $\mathcal{X}$, and let $\mathcal{Y}^{\mathcal{X}}$ denote all possible mappings from $\mathcal{X}$ to $\mathcal{Y}$. For an integer $n$, denote $[n] := \{1, 2, \ldots, n\}$.

For simplicity of exposition, we will focus on the setting where $\mathcal{Y}$ is binary throughout, i.e. $\mathcal{Y} = \{-1, 1\}$. We note that our techniques are more general, however, and may be adapted to the case of finite, multi-class action spaces (see Appendix C for details).

### 2.2 Online Multi-Group Learning

We formally define the multi-group learning model as follows. Let $\mathcal{G} \subset 2^{\mathcal{X}}$ be a collection of (possibly non-disjoint) groups, and let $\mathcal{H} \subset \mathcal{Y}^{\mathcal{X}}$ be a *hypothesis class* of functions $h : \mathcal{X} \to \mathcal{Y}$ mapping from contexts to actions. Let $\ell : \mathcal{Y} \times \mathcal{Y} \to [0, 1]$ be a bounded loss function. In each round $t \in [T]$:

1. Nature chooses $(x_t, y_t) \in \mathcal{X} \times \mathcal{Y}$ and reveals $x_t$.
2. The learner chooses an action $\hat{y}_t \in \mathcal{Y}$.
3. Nature reveals $y_t \in \mathcal{Y}$.
4. The learner incurs loss $\ell(\hat{y}_t, y_t) \in [0, 1]$.

The choices of Nature and the learner may be randomized. In the standard online prediction setting, the regret of the learner is the difference between the cumulative loss of the learner and that of the best-in-hindsight hypothesis from $\mathcal{H}$: $\text{Reg}_T(\mathcal{H}) := \sum_{t=1}^{T} \ell(\hat{y}_t, y_t) - \min_{h \in \mathcal{H}} \sum_{t=1}^{T} \ell(h(x_t), y_t)$. The goal of the learner is to achieve sublinear (in $T$) expected regret.

In multi-group online learning, we consider the regret of the learner on subsequences of rounds $(t \in [T] : x_t \in g)$ defined by the groups $g \in \mathcal{G}$ and the sequence of contexts $x_1, \ldots, x_T$. Specifically, the *(multi-group) regret of the learner on group $g$* is

$$\text{Reg}_T(\mathcal{H}, g) := \sum_{t=1}^{T} g(x_t)\ell(\hat{y}_t, y_t) - \min_{h \in \mathcal{H}} \sum_{t=1}^{T} g(x_t)\ell(h(x_t), y_t). \tag{1}$$

Crucially, the best hypothesis for one group may differ from that of another group. Further, groups may intersect, precluding the strategy of simply running a separate no-regret algorithm for each group. The learner seeks to achieve achieve sublinear expected regret, on all groups $g \in \mathcal{G}$ simultaneously.

## 2.3 Group Oracle-Efficiency

The main challenge posed in this work is to design computationally efficient algorithms that work with both large hypothesis classes $\mathcal{H}$ and large collections of groups $\mathcal{G}$. The prior work of [Ach+23] shows how to use the following optimization oracle to avoid explicitly enumerating the hypothesis class $\mathcal{H}$ (but still require enumerating $\mathcal{G}$).

**Definition 2.1** (Optimization Oracle). *For some error parameter $\alpha \geq 0$ and function class $\mathcal{F} \in \overline{\mathcal{Z}}^{\mathcal{X}}$, an $\alpha$-approximate optimization oracle OPT takes a collection of pairs $(x_1, z_1), \ldots, (x_m, z_m) \in \mathcal{X} \times \mathcal{Z}$, a sequence of weights $w_1, \ldots, w_m \in \mathbb{R}$, and a sequence of $m$ loss functions $\ell_i : \overline{\mathcal{Z}} \times \mathcal{Z} \to [-1, 1]$ and outputs a function $\hat{f} := \mathrm{OPT}(\{(x_i, z_i, w_i)\}_{i=1}^m) \in \mathcal{F}$ satisfying:*

$$\sum_{i=1}^m w_i \ell_i(\hat{f}(x_i), z_i) \leq \inf_{f \in \mathcal{F}} \sum_{i=1}^m w_i \ell_i(f(x_i), z_i) + \alpha.$$

Instantiating (in Definition 2.1) $\mathcal{Z}$ as our action space $\mathcal{Y}$, each $\ell_i$ as the given loss $\ell$ of our problem, and $\mathcal{F}$ as our hypothesis class $\mathcal{H}$ gives a standard empirical risk minimization (ERM) oracle over a dataset $\{(x_i, y_i)\}_{i=1}^m$. We present this more general definition to distinguish the action space $\mathcal{Y}$ of the problem from the output space of the oracle (see Definition 2.2).

The optimization oracle is regarded as a natural computational primitive because, for many problems in machine learning, various heuristic methods (e.g., stochastic gradient descent) appear to routinely solve such problem instances despite the worst-case intractability of such problems.

The work of [Ach+23] still relies on explicit enumeration of $\mathcal{G}$. We show how this can be avoided using a joint optimization oracle for $\mathcal{G} \times \mathcal{H}$, defined as follows.

**Definition 2.2** (($\mathcal{G}, \mathcal{H}$)-optimization oracle). *Fix an error parameter $\alpha \geq 0$. For a collection of groups $\mathcal{G} \in 2^{\mathcal{X}}$, a collection of hypotheses $\mathcal{H} \subseteq \mathcal{Y}^{\mathcal{X}}$, and a sequence of $m$ loss functions $\ell_i : (\{0, 1\} \times \mathcal{Y}) \times (\mathcal{Y} \times \mathcal{Y}) \to [-1, 1]$, an $\alpha$-approximate $(\mathcal{G}, \mathcal{H})$-optimization oracle $\mathrm{OPT}_{(\mathcal{G}, \mathcal{H})}^{\alpha}$ is an $\alpha$-approximation optimization oracle (Definition 2.1) that outputs a pair $(\tilde{g}, \tilde{h}) \in \mathcal{G} \times \mathcal{H}$ satisfying:*

$$\sum_{i=1}^m w_i \ell_i((\tilde{g}(x_i), \tilde{h}(x_i)), (y_i, y_i')) \geq \sup_{(g^*, h^*) \in \mathcal{G} \times \mathcal{H}} \sum_{i=1}^m w_i \ell_i((g^*(x_i), h^*(x_i)), (y_i, y_i')) - \alpha. \quad (2)$$

If $\mathcal{Y} = \{-1, 1\}$ (which we assume in the main paper body), this oracle outputs a group-hypothesis pair $(\tilde{g}, \tilde{h})$ that maximizes the batch loss over $m$ examples $(x_i, (y_i, y_i'))$ in $\mathcal{X} \times \{-1, 1\}^2$. [GKR22] also made such an assumption and gave two implementations: one based on cost-sensitive classification oracles for $\mathcal{G}$ and $\mathcal{H}$ separately, the other a heuristic algorithm that is empirically effective. Details of these oracle instantiations are included in Appendix B.3.

We also require an optimization oracle for $\mathcal{H}$ itself, defined similarly. This can be thought of simply as (exact) empirical risk minimization over $\mathcal{H}$.[1]

**Definition 2.3** ($\mathcal{H}$-optimization oracle). *For a collection of hypotheses $\mathcal{H} \subseteq \mathcal{Y}^{\mathcal{X}}$, and a sequence of $m$ loss functions $\ell_i : \mathcal{Y} \times \mathcal{Y} \to [-1, 1]$, an $\mathcal{H}$-optimization oracle $\mathrm{OPT}_{\mathcal{H}}$ is a $0$-approximation optimization oracle (Definition 2.1, with $\alpha = 0$) that outputs a hypothesis $h \in \mathcal{H}$ satisfying $\sum_{i=1}^m w_i \ell_i(h(x_i), y_i) \leq \inf_{h^* \in \mathcal{H}} \sum_{i=1}^m w_i \ell_i(h^*(x_i), y_i).$*

# 3 Warm-up: I.I.D. Setting

In this section, as a warm-up, we consider a setting where Nature is stochastic and oblivious: the $(x_t, y_t)$ are drawn i.i.d. from a single fixed (but unknown) distribution $\mu$, independent of any choices of the learner. In a standard online prediction setting with i.i.d. data, it suffices to use a "follow-the-leader" (FTL) strategy (see, e.g., [Haz22]), which can be easily implemented using an optimization oracle for $\mathcal{H}$. However, such a strategy only guarantees low regret on $g = \mathcal{X}$. To achieve low regret on all (possibly intersecting) groups $g \in \mathcal{G}$ simultaneously, we need a multi-group analogue of FTL.

---

[1]In fact, it will suffice to have a substantially simpler oracle that, given a single $(x, y) \in \mathcal{X} \times \mathcal{Y}$, determines if there exists $h \in \mathcal{H}$ such that $h(x) = y$.

What makes FTL work in the standard online prediction setting is the instantaneous expected regret bound of empirical risk minimization (ERM) on i.i.d. data. Therefore, it is natural to replace ERM with a batch multi-group algorithm [TH22; GKR22]; this will ensure the requisite instantaneous guarantee on all groups $g \in \mathcal{G}$. We show how to use the oracle-efficient algorithm LISTUPDATE of [GKR22] for the online multi-group problem.

Throughout this section, we assume $\ell$ is the zero-one loss (for simplicity), and that $\mathcal{H}$ and $\mathcal{G}$ both have VC dimension at most $d \geq 1$. For any $g \in \mathcal{G}$, let $P(g) := \mathbb{E}_{(x,y) \sim \mu}[g(x)]$ be the probability mass of group $g$.

**Theorem 3.1** (Theorem 16 of [GKR22])**.** *For any $\delta \in (0,1)$, given $n$ i.i.d. training samples $\{(x_i, y_i)\}_{i=1}^n$ from $\mu$, the* LISTUPDATE *algorithm[2] returns a function $f : \mathcal{X} \to \mathcal{Y}$ such that, with probability $1 - \delta$, for any group $g \in \mathcal{G}$,*

$$\mathbb{E}[\ell(f(x), y) \mid x \in g] \leq \min_{h \in \mathcal{H}} \mathbb{E}[\ell(h(x), y) \mid x \in g] + \frac{1}{P(g)} \cdot O\left(\left(\frac{d \log n + \log(1/\delta)}{n}\right)^{1/3}\right).$$

*Moreover,* LISTUPDATE *makes* $\mathrm{poly}(n, d, \log(1/\delta))$ *calls to a $(\mathcal{G}, \mathcal{H})$ optimization oracle.*

Our algorithm, ONLINE LISTUPDATE, forms its prediction $\hat{y}_t$ in round $t$ as follows:

- Run LISTUPDATE on the samples from previous rounds $(x_1, y_1), \ldots, (x_{t-1}, y_{t-1})$.
- Let $f_t$ denote the function returned by LISTUPDATE, and predict $\hat{y}_t := f_t(x_t)$.

Using Theorem 3.1, we obtain the following multi-group regret bound for ONLINE LISTUPDATE.

**Theorem 3.2.** *If $(x_t, y_t)$ are drawn i.i.d. from a fixed distribution $\mu$ over $\mathcal{X} \times \mathcal{Y}$,* ONLINE LISTUP-DATE *achieves the following expected multi-group regret bound: for all $g \in \mathcal{G}$,*

$$\mathbb{E}[\mathrm{Reg}_T(\mathcal{H}, g)] = O\left((d \log T)^{1/3} T^{2/3} + \sqrt{dT \log T}\right).$$

The proof of Theorem 3.2 is given in Appendix A.

## 4 Group Oracle-Efficiency with Smooth Contexts

In this section, we first describe a natural problem setting in which oracle-efficient online multi-group learning is possible: the smoothed online learning setting (Section 4.1), for which the i.i.d. setting of Section 3 is a special case. We then present our main algorithm, Algorithm 1, for achieving oracle-efficient online multi-group learning (Section 4.2). Easy modifications of this main algorithm will admit oracle-efficient online multi-group learning for other common online learning specifications, as described in Section 5.

### 4.1 Smoothed Online Learning

We now describe *smoothed online learning*, a prevalent model in recent literature in computation-ally efficient online learning that formalizes the natural relaxation that Nature is not maximally adversarial [RST11; HRS22; Hag+22; Blo+22]. The main assumption is that, instead of choosing *arbitrary* (possibly worst-case) examples $(x_t, y_t) \in \mathcal{X} \times \mathcal{Y}$ at every round, Nature adversarially fixes a distribution $\mu_t$ over $\mathcal{X}$ and draws $x_t \sim \mu_t$, while still drawing $y_t$ adversarially. Formally, we restrict such distributions to be $\sigma$-*smooth*, following the definitions of [Blo+22; HRS22].

**Definition 4.1** ($\sigma$-smooth distribution)**.** *Let $\mu$ be some probability measure on $\mathcal{X}$, and let $\mathcal{B}$ be a base measure on $\mathcal{X}$. The distribution $\mu$ is $\sigma$-smooth (with respect to $\mathcal{B}$) if $\mu$ is absolutely continuous[3] with respect to $\mathcal{B}$ and*

$$\mathrm{ess} \sup \frac{d\mu}{d\mathcal{B}} \leq \frac{1}{\sigma}.$$

*We denote the set of all $\sigma$-smooth distributions on $\mathcal{X}$ with respect to the measure $\mathcal{B}$ as $\mathcal{S}_\sigma(\mathcal{X}, \mathcal{B})$. If $\mathcal{B}$ is clear from context, we simply write $\mathcal{S}_\sigma(\mathcal{X})$. We assume that we have sample access to $\mathcal{B}$ throughout. For simplicity, one may assume $\mathcal{B}$ is uniform on $\mathcal{X}$.*

---

[2]Technically, we use the TRAINBYOPT variant of LISTUPDATE from Theorem 16 of [GKR22].

[3]A probability measure $\mu$ is *absolutely continuous* to another measure $\mathcal{B}$ if, for every $\mathcal{B}$-measurable set $A$, $\mathcal{B}(A) = 0$ implies $\mu(A) = 0$.

Definition 4.1 interpolates between the benign setting where $x_t$ are drawn i.i.d. from $\mu$ when $\sigma = 1$, and the fully adversarial setting when $\sigma$ approaches $0$. In this sense, the warm-up result of Section 3 is a special case of this setting when $\sigma = 1$ and $\mu$ is fixed for all rounds. Note that this definition does not restrict the choice of $y_t$ at all; $y_t$ may still be chosen adversarially.

With this definition in hand, consider the following specification of the learning game outlined in Section 2.2, henceforth refered to as the *$\sigma$-smooth online learning setting*. For each round $t \in [T]$:

1. Nature fixes a distribution $\mu_t \in \mathcal{S}_\sigma(\mathcal{X})$ that may depend in any way on the entire history prior to round $t$. Nature samples $x_t \sim \mu_t$ and chooses $y_t \in \mathcal{Y}$ adversarially; $x_t$ is revealed to the learner.
2. The learner (randomly) chooses an action $\hat{y}_t \in \mathcal{Y}$.
3. Nature reveals $y_t \in \mathcal{Y}$, and the learner incurs the loss $\ell(\hat{y}_t, y_t) \in [0, 1]$.

We now depart from the previous literature that considers oracle-efficient algorithms in this setting ([HRS22; Blo+22]), as we focus on the more difficult objective of minimizing multi-group regret over a collection $\mathcal{G}$, as in Equation (1). This setting will allow us to employ our $(\mathcal{G}, \mathcal{H})$-optimization oracle in Definition 2.2; a full description of our algorithm is now in order in Section 4.2.

## 4.2 Algorithm for Smooth Contexts

In this section, we present Algorithm 1, our main algorithm for multi-group online learning for the $\sigma$-smooth setting. At a high level, our algorithm takes inspiration from the very general *adversary-moves-first (AMF) framework* for multiobjective online learning of [Lee+22]. Our algorithm can be thought of as a sequential game between two competing players: an adversarial $(\mathcal{G}, \mathcal{H})$-player and the learner, referred to, in the context of Algorithm 1 as the $\mathcal{H}$-player. On each round $t$, the $(\mathcal{G}, \mathcal{H})$-player employs a $(\mathcal{G}, \mathcal{H})$-optimization oracle, $\mathrm{OPT}^\alpha_{(\mathcal{G}, \mathcal{H})}$, to play a distribution over group-hypothesis pairs that maximizes the misfortune of the $\mathcal{H}$-player, based on the history up until $t$. Upon receiving this distribution (and the context $x_t$ from Nature), the $\mathcal{H}$-player chooses $\hat{y}_t$ randomly according to a distribution obtained by solving a simple constant-size linear program, and then incurs the loss $\ell(\hat{y}_t, y_t)$. The $(\mathcal{G}, \mathcal{H})$-player, taking this new loss into account, can now adjust his strategy to foil the $\mathcal{H}$-player in the next round by putting mass on the groups on which the $\mathcal{H}$-player performs poorly. Crucially, neither $\mathcal{G}$ nor $\mathcal{H}$ is ever accessed directly, although our proofs need to maintain a distribution over $\mathcal{G} \times \mathcal{H}$. In order to do this, we make the crucial observation that FTPL maintains an *implicit* distribution over $\mathcal{G} \times \mathcal{H}$ and we sparsely approximate that distribution through repeatedly querying the $\mathrm{OPT}^\alpha_{(\mathcal{G}, \mathcal{H})}$ oracle. (For clarity, we use the tilde decoration, $\tilde{h}$ and $\tilde{g}$, on hypotheses and groups obtained by the $(\mathcal{G}, \mathcal{H})$-player using $\mathrm{OPT}^\alpha_{(\mathcal{G}, \mathcal{H})}$.)

**Main algorithm.** For any $x \in \mathcal{X}$, define $\tilde{\ell}_x : (\{0, 1\} \times \mathcal{Y}) \times (\mathcal{Y} \times \mathcal{Y}) \to [-1, 1]$ as:

$$\tilde{\ell}_x((\tilde{g}, \tilde{h}), (y', y)) := \tilde{g}(x) \left( \ell(y', y) - \ell(\tilde{h}(x), y) \right), \tag{3}$$

where $\ell(\cdot, \cdot)$ is the loss given by the learning problem. The quantity $\tilde{\ell}_x$ is the loss that the $(\mathcal{G}, \mathcal{H})$-player is maximizing; it corresponds to the single-round regret of the learner on group $g$ to the hypothesis $h$ if the context on that round is $x$.

The $(\mathcal{G}, \mathcal{H})$-player will employ the FTPL style strategy of [Blo+22], adapted to our setting. For each round $t$, this requires generating $n$ *perturbation examples* as extra input to $\mathrm{OPT}^\alpha_{(\mathcal{G}, \mathcal{H})}$. To generate these hallucinated perturbation examples, we independently draw $z_{t,j} \sim \mathcal{B}$ and $\gamma_{t,j} \sim N(0, 1)$, samples $j \in [n]$ from the base measure and the standard Gaussian, respectively. In this section, $\mathcal{Y} = \{-1, 1\}$ and $\mathcal{H} \subseteq \{-1, 1\}^\mathcal{X}$, so we use the perturbations in their FTPL variant for binary-valued action spaces (outlined in Appendix B.6),

$$\pi^{\mathrm{bin}}_{t,n}(g, h, \eta) := \sum_{j=1}^n \frac{\eta \gamma_{t,j} g(z_{t,j}) h(z_{t,j})}{\sqrt{n}}. \tag{4}$$

**Remark.** *We focus on the setting where $\mathcal{Y} = \{-1, 1\}$ for ease of exposition, but settings in which $\mathcal{Y}$ is a general finite set can be handled with easy modifications. See Appendix C for details.*

**Remark.** *Multi-group online learning settings other than the $\sigma$-smooth setting can be handled appropriately simply by replacing the strategy of the $(\mathcal{G}, \mathcal{H})$-player by an appropriate no-regret*

**Algorithm 1** Algorithm for Group-wise Oracle Efficiency (for smoothed online learning)

---

**Input:** Perturbation strength $\eta > 0$; perturbation count $n \in \mathbb{N}$; number of oracle calls $M \in \mathbb{N}$.

1: **for** $t = 1, 2, 3, \ldots, T$ **do**
2:      Receive a context $x_t \sim \mu_t$ from Nature.
3:      **for** $i = 1, 2, 3, \ldots, M$ **do**
4:         $(\mathcal{G}, \mathcal{H})$-**player:** Draw $n$ hallucinated examples as in Equation (4) to construct $\pi_{t,n}^{\mathrm{bin}}$.
5:         $(\mathcal{G}, \mathcal{H})$-**player:** Using the entire history $\{(\hat{y}_s, y_s)\}_{s=1}^{t-1}$ so far, call $\mathrm{OPT}_{(\mathcal{G}, \mathcal{H})}^{\alpha}$ to obtain $(\tilde{g}_t^{(i)}, \tilde{h}_t^{(i)}) \in \mathcal{G} \times \mathcal{H}$ satisfying:

$$\sum_{s=1}^{t-1} \tilde{\ell}_{x_s}((\tilde{g}_t^{(i)}, \tilde{h}_t^{(i)}), (\hat{y}_s, y_s)) + \pi_{t,n}^{\mathrm{bin}}(\tilde{g}_t^{(i)}, \tilde{h}_t^{(i)}, \eta)$$

$$\geq \sup_{(g^*, h^*) \in \mathcal{G} \times \mathcal{H}} \sum_{s=1}^{t-1} \tilde{\ell}_{x_s}((g^*, h^*), (\hat{y}_s, y_s)) + \pi_{t,n}^{\mathrm{bin}}(g^*, h^*, \eta) - \alpha \quad (5)$$

6:      **end for**
7:      $\mathcal{H}$-**player:** Call $\mathrm{OPT}_{\mathcal{H}}$ twice on the singleton datasets $\{(x_t, 1)\}$ and $\{(x_t, -1)\}$, with the 0-1 loss, obtaining:

$$h_1' \in \underset{h^* \in \mathcal{H}}{\arg\min} \, \mathbf{1} \{h^*(x_t) \neq 1\}, \quad h_{-1}' \in \underset{h^* \in \mathcal{H}}{\arg\min} \, \mathbf{1} \{h^*(x_t) \neq -1\}.$$

8:      $\mathcal{H}$-**player:** Solve the linear program

$$\min_{p, \lambda \in \mathbb{R}} \quad \lambda$$

$$\text{subj. to} \quad \sum_{i=1}^{M} p \tilde{\ell}_{x_t}((\tilde{g}_t^{(i)}, \tilde{h}_t^{(i)}), (h_1'(x_t), y)) + (1-p)\tilde{\ell}_{x_t}((\tilde{g}_t^{(i)}, \tilde{h}_t^{(i)}), (h_{-1}'(x_t), y)) \leq \lambda$$

$$\forall y \in \{-1, 1\}$$

$$0 \leq p \leq 1.$$

9:      Sample $b \sim \mathrm{Ber}(p)$ where $b \in \{-1, 1\}$, let $h_t = h_b'$.
10:     Learner commits to the action $\hat{y}_t = h_t(x_t)$; Nature reveals $y_t$.
11:     Learner incurs the loss $\ell(\hat{y}_t, y_t)$.
12: **end for**

---

*algorithm with access to* $\mathrm{OPT}_{(\mathcal{G}, \mathcal{H})}^{\alpha}$. *Examples of such variants are given in Section 5, and the general framework for such modifications is given in Appendix B.*

**Remark.** *With appropriate modifications, one can instantiate the $(\mathcal{G}, \mathcal{H})$-player with the FTPL style strategy of [Hag+22] instead, inheriting the $\sigma^{-1/4}$ dependence summarized in Table 1. Our focus in this paper is not on the dependence on $\sigma$, however, so our main exposition centers around the similar algorithmic techniques of [Blo+22].*

It is clear that $\mathcal{G}$ and $\mathcal{H}$ are never accessed except through $\mathrm{OPT}_{(\mathcal{G}, \mathcal{H})}^{\alpha}$ and $\mathrm{OPT}_{\mathcal{H}}$. We make $M$ oracle calls to $\mathrm{OPT}_{(\mathcal{G}, \mathcal{H})}^{\alpha}$ and two oracle calls to $\mathrm{OPT}_{\mathcal{H}}$ at each round.

**Theorem 4.1.** *Let $\mathcal{Y} = \{-1, 1\}$ be a binary action space, $\mathcal{H} \subseteq \{-1, 1\}^{\mathcal{X}}$ be a binary-valued hypothesis class, $\mathcal{G} \subseteq 2^{\mathcal{X}}$ be a (possibly infinite) collection of groups, and $\ell : \{-1, 1\} \times \{-1, 1\} \to [0, 1]$ be a bounded loss function. Let the VC dimensions of $\mathcal{H}$ and $\mathcal{G}$ both be bounded by $d$. Let $\alpha \geq 0$ be the approximation error of the oracle $\mathrm{OPT}_{(\mathcal{G}, \mathcal{H})}^{\alpha}$. If we are in the $\sigma$-smooth online learning setting, then, for $M = \mathrm{poly}(T), n = \mathrm{poly}(T/\sigma)$, and $\eta = \mathrm{poly}(T/\sigma)$, Algorithm 1 achieves, for each $g \in \mathcal{G}$:*

$$\mathbb{E}[\mathrm{Reg}_T(\mathcal{H}, g)] \leq O\left(\sqrt{\frac{dT \log T}{\sigma}} + \alpha T\right),$$

*where the expectation is over all the randomness of the $(\mathcal{G}, \mathcal{H})$-player's perturbations and the $\mathcal{H}$-player's Bernoulli choices. (See Corollary B.7.1 for precise settings of $M$, $n$, and $\eta$.)*

**Technical details.** We solve three main difficulties toward ensuring that our algorithm achieves diminishing multi-group regret while maintaining computational efficiency for large or infinite $\mathcal{G}$, which we outline here. The full proof of Theorem 4.1 is in Appendix B.

First, although the general framework of casting online learning problems with multiple objectives as two-player games is not new ([Lee+22; HJZ24; HPY23]), previous works have employed a multiplicative weights algorithm to hedge against the multiple objectives, requiring explicit enumeration. Departing from previous literature, however, our $(\mathcal{G}, \mathcal{H})$-player uses a *follow-the-perturbed leader (FTPL)* style algorithm (see, e.g., [KV05]) with $\mathrm{OPT}_{(\mathcal{G}, \mathcal{H})}$. The particular follow-the-perturbed leader variant of [Blo+22] constructs "perturbations" via a set of fake examples drawn from the the base measure $\mathcal{B}$ on $\mathcal{X}$, and, is thus suitable for our oracle and problem setting.

Second, a key property needed by the proof of Algorithm 1 is that the $\mathcal{H}$-player must receive a *distribution* over $\mathcal{G} \times \mathcal{H}$ to reduce the complex multi-objective criterion of performing well against all $(g, h) \in \mathcal{G} \times \mathcal{H}$ to a scalar quantity. Previous work directly supplied this distribution through the multiplicative weights algorithm. However, this would involve explicitly enumerating $\mathcal{G}$ and $\mathcal{H}$. On the other hand, using an FTPL algorithm as is would only output a *single* action from $\mathcal{G} \times \mathcal{H}$, which is insufficient. To remedy this, we make the crucial observation that FTPL algorithms *implicitly* maintain a distribution over $\mathcal{G} \times \mathcal{H}$ through the randomness of their perturbations, and, thus, we construct the empirical approximation of this distribution through repeatedly calling $\mathrm{OPT}_{(\mathcal{G}, \mathcal{H})}$. Standard uniform convergence arguments are used to bound the number of oracle calls needed. An argument employing the minimax theorem shows that the final regret guarantee of the entire algorithm essentially inherits the regret of the FTPL algorithm, plus sublinear error terms.

Finally, the $\mathcal{H}$-player chooses a distribution over $\mathcal{Y}$ by by solving an exceedingly simple linear program (LP) with two optimization variables, $p$ and $\lambda$. The value $p \in [0, 1]$ corresponds to the parameter of a Bernoulli distribution from which we sample to choose $\hat{y}_t$. This choice of action corresponds exactly to choosing the minimax optimal strategy against the worst-case $y$ that Nature could select. We employ similar techniques as [Lee+22], analyzing the value of this min-max game as if Nature (the $\max$ in the min-max) had gone first instead. The two calls to $\mathrm{OPT}_{\mathcal{H}}$ are used just to find the actions achievable by $\mathcal{H}$ on $x_t$. (Note that it is possible that $h'_{y'}(x_t) \neq y'$ for some $y' \in \mathcal{Y}$, in which case the Learner will always play $-y'$, regardless of the value of $p$.)

## 5 Group Oracle-Efficiency in Other Settings

In the previous section, we presented an algorithm that achieves $o(T)$ expected regret for all $g \in \mathcal{G}$, satisfying our main desideratum from Section 2.2. However, in some cases, we may want something more. Suppose that some groups are rarer than others; in this case, a natural extension would be to ensure a stronger "adaptive" regret bound that instead scales with $T_g := \sum_{t=1}^{T} g(x_t)$, the number of times group $g$ appeared in the $T$ rounds. We note that the algorithm of [BL20] achieves such a multi-group regret guarantee (for finite $\mathcal{H}$ and $\mathcal{G}$), but their algorithm is not oracle-efficient. So a question that remains is whether such guarantees can be achieved in an oracle-efficient manner.

In this section, we are back in the general fully adversarial multi-group online learning setting of Section 2.2 (without i.i.d. or smoothness assumptions). We discuss how to modify Algorithm 1 via the *Generalized Follow-the Perturbed-Leader (GFTPL) framework* of [Dud20; Wan+22] to obtain regret guarantees on group $g$ where the dependence on $T$ is replaced (at least in part) with $T_g$. Due to space limitations, we give a sketch here; the full details are in Appendix C.

We first make a simple observation that motivates our use of more advanced oracle-efficient online learning techniques. Recall the $\tilde{g}$-specific per-round regret to $\tilde{h}$ of playing $h(x_t)$ at round $t \in [T]$:

$$\tilde{\ell}_{x_t}((\tilde{g}, \tilde{h}), (h(x_t), y)) = \tilde{g}(x_t) \left( \ell(h(x_t), y) - \ell(\tilde{h}(x_t), y) \right).$$

The job of the $(\mathcal{G}, \mathcal{H})$-player is to run a no-regret algorithm to maximize this quantity in aggregate, as described in Section 4.2 and detailed in Appendix B.5. The online learning literature for *small-loss regret* focuses on developing algorithms that have regret depending on cumulative loss in hindsight instead of the number of rounds $T$ [HP05; CL06; GSV14]; this has the advantage of giving a tighter

regret bound when losses are small in magnitude. It is immediate that $\tilde{\ell}_{x_t}((\tilde{g}, \tilde{h}), (h(x_t), y)) = 0$ whenever $\tilde{g}(x_t) = 0$, so a small-loss regret would immediately give a $o(T_g)$ guarantee.

We focus on the case where $\mathcal{G}$ and $\mathcal{H}$ are finite, and $\mathcal{G} \times \mathcal{H}$ is the set of experts the $(\mathcal{G}, \mathcal{H})$-player has access to. Most small-loss regret algorithms would require prohibitive enumeration of $\mathcal{G} \times \mathcal{H}$ [HP05; CL06; GSV14; LS15], but the *GFTPL with small-loss bound algorithm* of [Wan+22] has the property that it is oracle-efficient *and* enjoys small-loss regret. This algorithm follows the GFTPL design template of [Dud+20], which, similar to the classic FTPL algorithm of [KV05], generates a noise vector to perturb each each decision of expert. However, whereas the classic FTPL algorithm generates $|\mathcal{G}| \times |\mathcal{H}|$ independent random noise variables, GFTPL only generates $N \ll |\mathcal{G}| \times |\mathcal{H}|$ independent random variables and uses a *perturbation matrix (PM)* $\Gamma \in [-1, 1]^{|\mathcal{G}||\mathcal{H}| \times N}$ to translate the noise vector back to $|\mathcal{G}| \times |\mathcal{H}|$ *dependent* perturbations.

The main challenge in instantiating a GFTPL algorithm is to construct a suitable $\Gamma$ for the problem at hand. [Wan+22] provide two sufficient conditions for $\Gamma$ that, respectively, imbue the GFTPL algorithm with oracle-efficiency and small-loss regret: *implementability* and *approximability*. In our setting, implementability requires that every column of $\Gamma$ correspond to a dataset of "fake examples" suitable to $\mathrm{OPT}_{(\mathcal{G},\mathcal{H})}^{\alpha}$. Approximability with parameter $\gamma > 0$ guarantees the stability property that the ratio $\mathbb{P}[(\tilde{g}_t, \tilde{h}_t) = (g, h)] / \mathbb{P}[(\tilde{g}_{t+1}, \tilde{h}_{t+1}) = (g, h)] \leq \exp(\gamma \eta_t)$ for all $(g, h) \in \mathcal{G} \times \mathcal{H}$, where $\eta_t > 0$ is the per-round learning rate of GFTPL. If such a $\Gamma$ exists, then instantiating the $(\mathcal{G}, \mathcal{H})$-player in Algorithm 1 with GFTPL (instead of the algorithm of [Blo+22]) gives us the stronger $o(T_g)$ regret guarantee. Full definitions and the proof, with the precise setting of $M$, can be found in Appendix C and Proposition C.3.1.

**Theorem 5.1.** *Assume $\mathcal{H}, \mathcal{G}$ are finite and there exists a $\gamma$-approximable and implementable perturbation matrix $\Gamma \in [-1, 1]^{|\mathcal{G}||\mathcal{H}| \times N}$. Let $\alpha \geq 0$ be the approximation parameter of $\mathrm{OPT}_{(\mathcal{G},\mathcal{H})}^{\alpha}$. Let the no-regret algorithm for the $(\mathcal{G}, \mathcal{H})$-player in Algorithm 1 be the GFTPL algorithm of [Wan+22] instantiated with $\Gamma$, with parameter $M = \mathrm{poly}(T)$. Then, for each $g \in \mathcal{G}$:*

$$\mathbb{E}[\mathrm{Reg}_T(\mathcal{H}, g)] \leq O\left(\sqrt{T_g} \max\left\{\gamma, \log |\mathcal{H}||\mathcal{G}|, \sqrt{N \log |\mathcal{H}||\mathcal{G}|}\right\} + \alpha T\right)$$

We give a particular setting in which one can easily construct an approximable and implementable $\Gamma$.

**Transductive Setting.** In the *transductive setting* of [SKS16; Dud+20], Nature reveals a set $X \subset \mathcal{X}$ to the Learner at the beginning of the learning process; then at each round $t \in [T]$, Nature can only choose $x_t$ from $X$. Let $N := |X|$ denote the number of different contexts that Nature chooses from. For this setting, we can explicitly construct $\Gamma$ to get the following result.

**Corollary 5.1.1** (Transductive setting)**.** *In the transductive setting, there exists a perturbation matrix $\Gamma \in [-1, 1]^{|\mathcal{G}||\mathcal{H}| \times 4N}$ such that Algorithm 1 with GFTPL parameterized with $M = \mathrm{poly}(T)$ and $\mathrm{OPT}_{(\mathcal{G},\mathcal{H})}^{\alpha}$ with error parameter $\alpha \geq 0$ satisfies:*

$$\mathbb{E}[\mathrm{Reg}_T(\mathcal{H}, g)] \leq O\left(\sqrt{T_g}\sqrt{\max\left\{\log |\mathcal{H}||\mathcal{G}|, \sqrt{N \log |\mathcal{H}||\mathcal{G}|}\right\}} + \alpha T\right) \quad \textit{for all } g \in \mathcal{G}.$$

Suppose that $N \leq T$ (which is the case in the transductive learning setting from [KK05]). Then the regret bound (ignoring the dependence on $\log(|\mathcal{H}||\mathcal{G}|)$) on group $g$ is $O(\sqrt{T_g} T^{1/4})$, which is asymptotically smaller than $\sqrt{T}$ whenever $T_g = o(\sqrt{T})$. If $N$ is fixed independent of $T$, then the regret bound is $O(\sqrt{T_g})$.

# 6    Conclusion and Future Work

In this paper, we design algorithms for online multi-group learning that are oracle-efficient and achieve diminishing $o(T)$ expected regret for all groups $g \in \mathcal{G}$ simultaneously, even when $\mathcal{G}$ is too large to explicitly enumerate. The most interesting future directions that we leave open in this work include designing oracle-efficient algorithms that achieve $o(T_g)$ group-specific regret for *infinite $\mathcal{H}$ and $\mathcal{G}$* and in more general settings.

**Acknowledgments**

We are grateful to Abhishek Shetty for pointing out possible improvements to the dependence on $\sigma$. We acknowledge funding support from a Google Faculty Research Award to Daniel Hsu, the NSF under grants IIS-2040971 and CCF-2008733, and the ONR under grants N00014-24-1-2700 and N00014-22-1-2713. Samuel Deng acknowledges funding from the Avanessians Doctoral Fellowship for Engineering Thought Leaders and Innovators in Data Science.

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

# A  Proof of Theorem 3.2

The goal is to prove that ONLINE LISTUPDATE satisfies, for all $g \in \mathcal{G}$,

$$\mathbb{E}[\text{Reg}_T(\mathcal{H}, g)] \leq O\left((d \log T)^{1/3} T^{2/3} + \sqrt{dT \log T}\right).$$

Fix any $g \in \mathcal{G}$. For each $h \in \mathcal{H}$, define

$$\text{Reg}_T(h, g) := \sum_{t=1}^{T} g(x_t)(\ell(\hat{y}_t, y_t) - \ell(h(x_t), y_t)).$$

By linearity of expectation, Theorem 3.1, and an elementary integral bound,

$$
\begin{aligned}
\mathbb{E}[\text{Reg}_T(h, g)] &= \sum_{t=1}^{T} \mathbb{E}[g(x_t)\ell(\hat{y}_t, y_t) - \mathbb{E}[g(x_t)\ell(h(x_t), y_t)] \\
&\leq 1 + \sum_{t=2}^{T} P(g) \cdot (\mathbb{E}[\ell(f_t(x_t), y_t) \mid x_t \in g] - \mathbb{E}[g(x_t)\ell(h(x_t), y_t) \mid x_t \in g]) \\
&\leq 1 + \sum_{t=2}^{T} \left((1 - \delta) \cdot O\left(\left(\frac{d \log(t-1) + \log(1/\delta)}{t-1}\right)^{1/3}\right) + \delta\right) \\
&= 1 + O\left((d \log T + \log(1/\delta))^{1/3} T^{2/3} + \delta T\right).
\end{aligned}
$$

Plug-in $\delta = 1/T$ to obtain

$$\mathbb{E}[\text{Reg}_T(h, g)] \leq O\left((d \log T)^{1/3} T^{2/3}\right).$$

It remains to relate $\max_{h \in \mathcal{H}} \mathbb{E}[\text{Reg}_T(h, g)]$ to $\mathbb{E}[\text{Reg}_T(\mathcal{H}, g)]$. Define

$$h_g \in \underset{h \in \mathcal{H}}{\arg\min} \, \mathbb{E}_{(x,y)\sim\mu}[g(x)\ell(h(x), y)] \quad \text{and} \quad \hat{h}_g \in \underset{h \in \mathcal{H}}{\arg\min} \sum_{t=1}^{T} g(x_t)\ell(h(x_t), y_t).$$

Then

$$\mathbb{E}[\text{Reg}_T(\mathcal{H}, g)] = \max_{h \in \mathcal{H}} \mathbb{E}[\text{Reg}_T(h, g)] + \mathbb{E}\left[\sum_{t=1}^{T} g(x_t)(\ell(h_g(x_t, y_t) - \ell(\hat{h}_g(x_t), y_t))\right].$$

Since $\mathcal{H}$ has VC dimension at most $d$, a standard uniform convergence argument implies

$$\mathbb{E}\left[\max_{h \in \mathcal{H}} T\mathbb{E}_{(x,y)\sim\mu}[g(x)\ell(h(x), y)] - \sum_{t=1}^{T} g(x_t)\ell(h(x_t), y_t)\right] \leq O\left(\sqrt{dT \log T}\right).$$

Using the definitions of $h_g$ and $\hat{h}_g$, we obtain

$$\mathbb{E}\left[\sum_{t=1}^{T} g(x_t)(\ell(h_g(x_t, y_t) - \ell(\hat{h}_g(x_t), y_t))\right] \leq O\left(\sqrt{dT \log T}\right).$$

Therefore, we conclude that

$$\mathbb{E}[\text{Reg}_T(\mathcal{H}, g)] \leq O\left((d \log T)^{1/3} T^{2/3} + \sqrt{dT \log T}\right).$$

This finishes the proof of Theorem 3.2. $\qquad\qquad\qquad\qquad\qquad\qquad\qquad\qquad$ $\square$

# B  Proof of Main Theorem 4.1

In this section, we prove the multi-group regret guarantee of our main algorithm, Algorithm 1. To restate the theorem, we aim to show, for all $g \in \mathcal{G}$,

$$\mathbb{E}[\mathrm{Reg}_T(\mathcal{H}, g)] \leq O\left(\sqrt{\frac{dT \log T}{\sigma}} + \alpha T\right).$$

More explicitly, we aim to show that:

$$\sum_{t=1}^{T} \mathbb{E}[g(x_t)\left(\ell(h_t(x_t), y_t) - \ell(h^*(x_t), y_t)\right)] \leq O\left(\sqrt{\frac{dT \log T}{\sigma}} + \alpha T\right),$$

where $h^* \in \min_{h \in \mathcal{H}} \sum_{t=1}^{T} g(x_t)\ell(h(x_t), y_t)$. We follow a generalization of the online minimax multiobjective optimization framework of [Lee+22], with techniques inspired by [HPY23].

## B.1  The AMF Algorithm Framework

We first restate their "adversary-moves-first" AMF algorithm of [Lee+22] and its main regret guarantee for convenience, as we will need to adapt and generalize it to our setting. Let $\mathcal{A}_t$ denote a general action space of the learner, and let $\mathcal{Z}_t$ denote the general action space of the adversary at round $t \in [T]$. In full generality, $\mathcal{A}_t$ and $\mathcal{Z}_t$ are allowed to change with the rounds $t \in [T]$. We differentiate this from the action space $\mathcal{Y}$ of the main body. For each round $t = 1, \ldots, T$, consider the following setting, which we refer to as the *multiobjective online optimization problem*:

1. The adversary selects a continuous, $d$-dimensional loss function $r_t : \mathcal{A}_t \times \mathcal{Z}_t \rightarrow [-1,1]^d$. Each component $r_t^j : \mathcal{A}_t \times \mathcal{Z}_t \rightarrow [-1,1]$ is convex in $\mathcal{A}_t$ and concave in $\mathcal{Z}_t$.
2. The learner selects an action $a_t \in \mathcal{A}_t$.
3. Nature observes the learner's action $a_t$ and responds with $z_t \in \mathcal{Z}_t$.
4. The learner incurs the $d$-dimensional loss $r_t(a_t, z_t)$.

In this setting, the learner's goal is to minimize the value of the maximum dimension of the accumulated loss vector after $T$ rounds:

$$\max_{j \in [d]} \sum_{t=1}^{T} r_t^j(a_t, z_t).$$

To benchmark the learner's performance, we consider the following quantity, which we refer to as the *adversary-moves-first (AMF) value at round* $t$.

**Definition B.1** (Adversary-Moves-First (AMF) Value at Round $t$,[Lee+22]). *The* adversary-moves-first (AMF) value at round $t$ *is the value:*

$$v_t^A := \max_{z_t \in \mathcal{Z}_t} \min_{a_t \in \mathcal{A}_t} \left(\max_{j \in [d]} r_t^j(a_t, z_t)\right). \tag{6}$$

We conceive of the value $v_t^A$ in (6) as the aspirational smallest value of the maximum coordinate of $r_t$ the learner could guarantee *if* the adversary had to reveal $z_t$ first and the learner could best respond with $a_t$. Per how the multiobjective online optimization problem is set up, however, the opposite is true — the learner must commit to an action $a_t \in \mathcal{A}_t$ first, and *then* the adversary is allowed to play $z_t \in \mathcal{Z}_t$ in response to maximize the learner's misfortune. Regardless, we can define a notion of regret with respect to this particular benchmark.

**Definition B.2** (Adversary-Moves-First (AMF) Regret). *Over $T$ rounds of the above* multiobjective online optimization problem*, the* adversary-moves-first regret *of the learner is:*

$$\mathrm{AMFReg}_T := \max_{j \in [d]} \left(\sum_{t=1}^{T} r_t^j(a_t, z_t) - v_t^A\right).$$

This notion measures the cumulative regret the learner has for not playing the action $a_t \in \mathcal{A}_t$ achieving the aspirational value $v_t^A$ at round $t$ over *all* $d$ coordinates of the loss vector.

**Algorithm 2** AMF Algorithm of [Lee+22]

---
1: **for** $t = 1, 2, 3, \ldots, T$ **do**
2:    Receive adversarially chosen action spaces $\mathcal{A}_t$ and $\mathcal{Z}_t$ and the $d$-dimensional loss function $r_t : \mathcal{A}_t \times \mathcal{Z}_t \to [-1, 1]^d$.
3:    Let
$$q_t^j := \frac{\exp\left(\eta \sum_{s=1}^{t-1} r_s^j(a_s, z_s)\right)}{\sum_{i \in [d]} \exp\left(\eta \sum_{s=1}^{t-1} r_s^i(a_s, z_s)\right)} \quad \text{for } j \in [d].$$

  For $t = 1$, let $q_t^j = 1/d$ for all $j \in [d]$.
4:    Solve the min-max optimization problem:
$$a_t \in \arg\min_{a \in \mathcal{A}_t} \max_{z \in \mathcal{Z}_t} \sum_{j \in [d]} q_t^j r_t^j(a, z). \tag{7}$$

5:    Commit to the action $a_t \in \mathcal{A}_t$, observe Nature's choice $z_t \in \mathcal{Z}_t$, and incur loss $r_t(a_t, z_t)$.
6: **end for**

---

A natural question, then, is to wonder if such a regret can be made to diminish sublinearly. That is, does there exist an algorithm such that $\mathrm{AMFReg}_T \leq o(T)$? [Lee+22] answer this in the affirmative, presenting Algorithm 2.

It is not immediately clear at first glance how Algorithm 2 translates to our multi-group online learning setting. In the next section, we will show a reduction from our setting to the this AMF framework. We will specialize Algorithm 2 to our setting and prove a "meta-theorem" similar to the original guarantee of Algorithm 2 and proceed to control the regret via that meta-theorem in subsequent sections.

## B.2 Reduction of multi-group online learning to AMF Framework

Recall that, in this paper, we actually care about the *online multi-group learning setting* described in Section 2.2. We restate the objective here for convenience.

In multi-group online learning, the learner has access to a hypothesis class $\mathcal{H}$ taking contexts from $\mathcal{X}$ and outputting actions in $\mathcal{Y}$, a common action space for the learner and Nature. There is a common loss function for the problem, $\ell(\cdot, \cdot) : \mathcal{Y} \times \mathcal{Y} \to [0, 1]$. To allay confusion, we refer to the adversary in the multi-group online setting of Section 2.2 as "Nature" and the adversary in the online multiobjective optimization problem of Section B.1 as "the adversary." The learners in both settings correspond to one another, so we just use "the learner." For clarity of exposition, we let $\mathcal{Y} = \{-1, 1\}$ be a binary action space for the remainder of Appendix B. The generalization to the case where $\mathcal{Y}$ takes $K$ discrete values is sketched in Appendix C.

We consider the regret of the learner on subsequences of rounds $(t \in [T] : x_t \in g)$ defined by the groups $g \in \mathcal{G}$ and the sequence of contexts $x_1, \ldots, x_T$. Specifically, the *(multi-group) regret of the learner on group $g$* is
$$\mathrm{Reg}_T(\mathcal{H}, g) := \sum_{t=1}^{T} g(x_t)\ell(\hat{y}_t, y_t) - \min_{h \in \mathcal{H}} \sum_{t=1}^{T} g(x_t)\ell(h(x_t), y_t). \tag{8}$$

Crucially, the best hypothesis for one group may differ from that of another group. The learner seeks to achieve achieve sublinear expected regret, on all groups $g \in \mathcal{G}$ simultaneously.

We now show a reduction from the multi-group online learning setting to the general multiobjective online optimization problem of the previous Section B.1. An important observation is that, when $\mathcal{Y} = \{-1, 1\}$, varying $h_t \in \mathcal{H}$ only affects the regret at round $t \in [T]$ insofar as its behavior on $x_t$. That is, for a fixed $x_t \in \mathcal{X}$ and $y \in \mathcal{Y}$, $\ell(h_t(x_t), y) \in \{\ell(-1, y), \ell(1, y)\}$. Note that it is possible for the set $\{h(x_t) : h \in \mathcal{H}\} \subseteq \mathcal{Y}$ to be a singleton set, in which case the learner will always take this unique action.

- Let $\mathcal{A}_t$, the learner's action space for round $t \in [T]$ in Section B.1, be the simplex $\Delta(\mathcal{Y})$.

- Let $\mathcal{Z}_t$, the adversary's action space in Section B.1, be $\mathcal{Z}_t = [0,1]$ for all $t \in [T]$, which will correspond to the parameter of a Bernoulli distribution over the binary-valued action space $\mathcal{Y}$ of Nature in the multi-group online learning problem.
- Let $r_t : \mathcal{A}_t \times \mathcal{Z}_t \to [-1,1]^d$, the adversarially chosen loss function, be $\mathcal{G} \times \mathcal{H}$-dimensional, with each coordinate corresponding to a pair $(\tilde{g}, \tilde{h}) \in \mathcal{G} \times \mathcal{H}$. Consider any $p, \gamma \in [0,1]$, each determining a Bernoulli distribution over $\mathcal{Y} = \{-1,1\}$. As in Section 4.2, for any $x \in \mathcal{X}$:

$$\tilde{\ell}_x((\tilde{g}, \tilde{h}), (y', y)) := \tilde{g}(x) \left( \ell(y', y) - \ell(\tilde{h}(x), y) \right).$$

Then, define:

$$r_t^{(\tilde{g}, \tilde{h})}(p, \gamma) := \mathbb{E}_{y' \sim p} \mathbb{E}_{y \sim \mathrm{Ber}(\gamma)} \left[ \tilde{\ell}_{x_t}((\tilde{g}, \tilde{h}), (y', y)) \right]. \tag{9}$$

Above, the loss $\ell(\cdot, \cdot)$ is the fixed loss of the multi-group online learning problem, and $x_t \in \mathcal{X}$ is the context chosen by Nature at round $t$, which indexes $r_t$.

It may now be slightly clearer how Algorithm 1 maps to Algorithm 2, but we provide a high-level overview here to prepare the reader for the subsequent sections.

- The distribution $q_t \in \Delta[d]$ in Line 3 of Algorithm 2 corresponds to the implicit distribution formed by querying $\mathrm{OPT}^\alpha_{(\mathcal{G}, \mathcal{H})}$ $M$ times to generate $\{(\tilde{g}_i, \tilde{h}_i)\}_{i=1}^M$, i.e. the $(\mathcal{G}, \mathcal{H})$-player.
- Solving the min-max optimization problem in Line 4, Equation (7) of Algorithm 2 corresponds to the two calls to the simple optimization problem solved by the $\mathcal{H}$-player and solving the simple one-dimensional linear program in Line 8 of Algorithm 1.

We make these correspondences formal in the subsequent sections. To organize this, we first formally show the correspondence between the $(\mathcal{G}, \mathcal{H})$-player and the construction of $q_t$, and the $\mathcal{H}$-player and Equation (7) in Algorithm 2.

## B.3 Instantiations of the $\mathrm{OPT}^\alpha_{(\mathcal{G}, \mathcal{H})}$ oracle

The computational primitive our algorithm assumes access to is a $(\mathcal{G}, \mathcal{H})$-optimization oracle, defined in Definition 2.2, and requoted here for ease of reference.

**Definition B.3** ($(\mathcal{G}, \mathcal{H})$-optimization oracle). *Fix an error parameter $\alpha \geq 0$. For a collection of groups $\mathcal{G} \in 2^{\mathcal{X}}$, a collection of hypotheses $\mathcal{H} \subseteq \mathcal{Y}^{\mathcal{X}}$, and a sequence of $m$ loss functions $\ell_i : (\{0,1\} \times \mathcal{Y}) \times (\mathcal{Y} \times \mathcal{Y}) \to [-1,1]$, an $\alpha$-approximate $(\mathcal{G}, \mathcal{H})$-optimization oracle $\mathrm{OPT}^\alpha_{(\mathcal{G}, \mathcal{H})}$ is an $\alpha$-approximation optimization oracle (Definition 2.1) that outputs a pair $(\tilde{g}, \tilde{h}) \in \mathcal{G} \times \mathcal{H}$ satisfying:*

$$\sum_{i=1}^m w_i \ell_i((\tilde{g}(x_i), \tilde{h}(x_i)), (y_i, y_i')) \geq \sup_{(g^*, h^*) \in \mathcal{G} \times \mathcal{H}} \sum_{i=1}^m w_i \ell_i((g^*(x_i), h^*(x_i)), (y_i, y_i')) - \alpha. \tag{10}$$

A common assumption in the literature on oracle-efficient online learning is positing the existence of some reasonable optimization oracle, typically commensurate to the ability to solve ERM. Although it is well-known that ERM is computationally hard in the worst-case, a bedrock of modern machine learning is the assumption that ERM is at least heuristically and approximately solvable. Although, for the purposes of our work, we assume access to this oracle as a black-box, it is natural to wonder if such an oracle can be instantiated. [GKR22] gives two such instantiations which we quote here for completeness.

We consider the specific instantiation of the $(\mathcal{G}, \mathcal{H})$-oracle in Algorithm 1 for a specific round $t \in [T]$, which aims to solve the following optimization problem for some $(\tilde{g}_t, \tilde{h}_t) \in \mathcal{G} \times \mathcal{H}$:

$$\sum_{s=1}^{t-1} \tilde{g}_t(x_s) \left( \ell(\hat{y}_s, y_s) - \ell(\tilde{h}_t(x_s), y_s) \right) \geq \mathrm{OPT} - \alpha, \tag{11}$$

where $\mathrm{OPT} := \sup_{g^*, h^* \in (\mathcal{G}, \mathcal{H})} \sum_{s=1}^{t-1} g^*(x_s) \left( \ell(\hat{y}_s, y_s) - \ell(h^*(x_s), y_s) \right)$.

In both instantiations of the oracle in [GKR22], the oracle aims to find a $(g, h) \in (\mathcal{G}, \mathcal{H})$ competitive to some reference model, $f : \mathcal{X} \to \{0,1\}$. For simplicity, as in the main body, we assume that

$\mathcal{Y} = \{0, 1\}$, and the "reference model" we compete with is given by the Learner's history of actions up to round $t$: $(x_1, \hat{y}_1), \ldots, (x_{t-1}, \hat{y}_{t-1})$. That is, we compare with the function $f : \mathcal{X} \to \{0, 1\}$ that maps $f(x_s) = \hat{y}_s$ for all $s = 1, \ldots, t-1$.

**Reduction to ternary classification.** The first instantiation of a $(\mathcal{G}, \mathcal{H})$ oracle in [GKR22] reduces the optimization oracle to the existence of a solver for a weighted ternary classification problem. The exposition here is quoted directly from [GKR22].

Start with a class $\mathcal{K}$ of *ternary* valued functions $p : \mathcal{X} \to \{0, 1, ?\}$. For each $p \in \mathcal{K}$, define the *p-derived group* and *p-derived hypothesis* as:

$$
g_p(x) = \begin{cases} 1 & \text{if } p(x) \in \{0, 1\} \\ 0 & \text{if } p(x) = ? \end{cases} \quad h_p(x) = \begin{cases} p(x) & \text{if } p(x) \in \{0, 1\} \\ 0 & \text{if } p(x) = ? \end{cases}
$$

This class $\mathcal{K}$ induces a set of pairs $(g_p, h_p)$ and a product class $(\mathcal{G}, \mathcal{H})_{\mathcal{K}} := \{(g_p, h_p) : p \in \mathcal{K}\}$. We may now define a *cost-sensitive classification* problem over $\mathcal{K}$ as follows, given an existing model $f : \mathcal{X} \to \{0, 1\}$, with the following costs:

$$
c_f((x, y), z) := \begin{cases} 0 & \text{if } z = ? \\ 1 & \text{if } f(x) = y \neq z \\ -1 & \text{if } z = y \neq f(x) \\ 0 & \text{otherwise} \end{cases}.
$$

For any distribution $\mu$ over $\mathcal{X} \times \{0, 1\}$, the associated cost-sensitive classification problem for costs $c_f((x, y), z)$ defined above is:

$$
p^* \in \arg\min_{p \in \mathcal{K}} \mathbb{E}_{(x,y) \sim \mu}[c_f((x, y), p(x))]. \tag{12}
$$

Many efficient algorithms that heuristically solve such optimization problems exist.

The main theorem from [GKR22], restated here, is the following:

**Theorem B.1.** *Fix any arbitrary distribution $\mu$ over $\mathcal{X} \times \{0, 1\}$. Let $\mathcal{K}$ be a class of ternary-valued functions $p : \mathcal{X} \to \{0, 1, ?\}$ and let $f : \mathcal{X} \to \{0, 1\}$ be any binary-valued model. Let $p^*$ be the solution to the cost-sensitive classification problem in Equation* (12)*. Then,*

$$
(g_p^*, h_p^*) \in \arg\max_{(g,h) \in (\mathcal{G}, \mathcal{H})_{\mathcal{K}}} \mathbb{E}_{(x,y) \sim \mu}[g(x)(\ell(f(x), y) - \ell(h(x), y))].
$$

*When $\mu$ is the empirical distribution over $x_1, \ldots, x_{t-1}$, the solution $(g_p^*, h_p^*)$ forms a solution to the optimization problem in Equation* (11) *when $(\mathcal{G}, \mathcal{H})_{\mathcal{K}} = \mathcal{G} \times \mathcal{H}$.*

We refer the reader to Section 4.2 in [GKR22] for a proof.

**Reduction to alternating maximization.** Another instantiation of the $(\mathcal{G}, \mathcal{H})$-oracle in [GKR22] is an alternating maximization approach. The reduction to ternary classification quoted above relies on an oracle for the class $\mathcal{K}$ and supplies guarantees for the derived class $(\mathcal{G}, \mathcal{H})_{\mathcal{K}}$. However, if we wish to begin with $\mathcal{G} \times \mathcal{H}$, we can take an "EM-style" alternating maximization approach that only requires ERM oracles for $\mathcal{G}$ and $\mathcal{H}$ separately. This approach only guarantees a saddle point local optimum.

The main idea is that, by holding $g$ fixed and solving for $h^*$, and vice versa, the following optimization problems are no harder than ERM over $\mathcal{G}$ and $\mathcal{H}$ individually:

$$
g^* \in \arg\max_{g^* \in \mathcal{G}} \mathbb{E}_{(x,y) \sim \mu}[g^*(x)(\ell(f(x), y) - \ell(h(x), y))] \tag{13}
$$

$$
h^* \in \arg\max_{h \in \mathcal{H}} \mathbb{E}_{(x,y) \sim \mu}[g(x)(\ell(f(x), y) - \ell(h^*(x), y))]. \tag{14}
$$

In Equation (13), $h$ is fixed and we solve for $g^*$; in Equation (14), $g$ is fixed, and we solve for $h^*$. With appropriate modifications to the distribution $\mu$ we can construct ERM problems for $g^*$ and $h^*$ comensurate to solving Equations (13) and (14). We refer the reader to Lemmas 21 and 22 in [GKR22] for the proofs.

This allows us to state an alternating maximization algorithm for finding a saddle point $(g, h) \in \mathcal{G} \times \mathcal{H}$ that gives a local optimum to Equation (11) in Algorithm 3. The corresponding theorem, restated from [GKR22] is:

**Theorem B.2.** *Let $\epsilon > 0$. Fix any empirical distribution over $(x_1, y_1), \ldots, (x_m, y_m)$, let $f : \mathcal{X} \to \{0, 1\}$ be an arbitrary model, and let $\mathcal{G}$ and $\mathcal{H}$ be arbitrary group and hypothesis classes. After solving at most $2/\epsilon$ ERM problems over each of $\mathcal{G}$ and $\mathcal{H}$ (in Equations (13) and (14), respectively), Algorithm 3 returns a pair $(g^*, h^*)$ with the properties that:*

1. *For every $h \in \mathcal{H}$,*
$$\sum_{i=1}^{m} g^*(x_i)\left(\ell(f(x_i), y_i) - \ell(h(x_i), y_i)\right) \leq \sum_{i=1}^{m} g^*(x_i)\left(\ell(f(x_i), y_i) - \ell(h^*(x_i), y_i)\right) + \epsilon.$$

2. *For every $g \in \mathcal{G}$,*
$$\sum_{i=1}^{m} g(x_i)\left(\ell(f(x_i), y_i) - \ell(h^*(x_i), y_i)\right) \leq \sum_{i=1}^{m} g^*(x_i)\left(\ell(f(x_i), y_i) - \ell(h^*(x_i), y_i)\right) + \epsilon.$$

---

**Algorithm 3** Alternating Maximization for $\mathcal{G} \times \mathcal{H}$ Oracle

---

**Input:** Dataset $\{(x_i, y_i)\}_{i=1}^{m}$, a model $f : \mathcal{X} \to \{0, 1\}$, error parameter $\epsilon$.
1: Initialize $(g^*, h^*) \in \mathcal{G} \times \mathcal{H}$ arbitrarily.
2: Let
$$\text{VAL} := \sum_{i=1}^{m} g^*(x_i)\left(\ell(f(x_i), y_i)) - \ell(h^*(x_i), y_i)\right)$$

3: Use ERM oracle for Equations (13) and (14) to solve for:

$$g^* \in \arg\max_{g \in \mathcal{G}} \sum_{i=1}^{m} g(x)\left(\ell(f(x), y) - \ell(h^*(x), y)\right)$$

$$h^* \in \arg\max_{h \in \mathcal{H}} \sum_{i=1}^{m} g^*(x)\left(\ell(f(x), y) - \ell(h(x), y)\right)$$

4: **while** $\sum_{i=1}^{m} g^*(x)\left(\ell(f(x), y) - \ell(h^*(x), y)\right) \geq \text{VAL} + \epsilon$ **do**
5:     Let
$$\text{VAL} := \sum_{i=1}^{m} g^*(x_i)\left(\ell(f(x_i), y_i)) - \ell(h^*(x_i), y_i)\right)$$

6:     Use ERM oracle for Equations (13) and (14) to solve for:

$$g^* \in \arg\max_{g \in \mathcal{G}} \sum_{i=1}^{m} g(x)\left(\ell(f(x), y) - \ell(h^*(x), y)\right)$$

$$h^* \in \arg\max_{h \in \mathcal{H}} \sum_{i=1}^{m} g^*(x)\left(\ell(f(x), y) - \ell(h(x), y)\right)$$

7: **end while**
8: Return $(g^*, h^*) \in \mathcal{G} \times \mathcal{H}$.

---

We believe that it is an interesting and worthwhile open question to develop more specific instantiations of this $(\mathcal{G}, \mathcal{H})$-oracle for more specific problem settings that are computationally efficient and have provable optimization guarantees.

### B.4 The group-hypothesis and hypothesis players

We formally define how the $(\mathcal{G}, \mathcal{H})$-player corresponds to the weights $q_t \in \Delta[d]$ in Algorithm 2. The crucial observation here is that the perturbations of the FTPL algorithm of [Blo+22] used in our setting form an implicit distribution over $\mathcal{G} \times \mathcal{H}$ that can be approximated by calling the $\text{OPT}_{(\mathcal{G},\mathcal{H})}^{\alpha}$ oracle $M$ times.

In the proceeding sections, we denote $\Delta(\mathcal{G} \times \mathcal{H})$ as the (possibly infinite-dimensional) space of measures over the functions $\mathcal{G} \times \mathcal{H}$. However, we will always only access sparse distributions on this

space, with a finite number of $(g, h)$ pairs in $\mathcal{G} \times \mathcal{H}$ obtaining nonzero mass. The following definition should make this clear.

**Definition B.4** (The distribution of the $(\mathcal{G}, \mathcal{H})$-player). *For any round $t \in [T]$, let $\{(\tilde{g}_i, \tilde{h}_i)\}_{i=1}^{M}$ be the $M$ samples drawn from querying $\mathrm{OPT}_{(\mathcal{G},\mathcal{H})}^{\alpha}$ in Algorithm 1. Let the empirical distribution $\tilde{q}_t \in \Delta(\mathcal{G} \times \mathcal{H})$ be the distribution of the $(\mathcal{G}, \mathcal{H})$-player at round $t$.*

It is easy to see that $\tilde{q}_t$ is a valid distribution over $(\mathcal{G}, \mathcal{H})$, with measure over $A \subseteq \mathcal{G} \times \mathcal{H}$, defined by:

$$P_M(A) := \frac{1}{M} \sum_{i=1}^{M} \delta_{(\tilde{g}_i, \tilde{h}_i)}(A),$$

where $\delta_{(\tilde{g}_i, \tilde{h}_i)}$ is the Dirac measure of $(\tilde{g}_i, \tilde{h}_i)$ falling into the set $A$. The stochasticity of $(\tilde{g}_i, \tilde{h}_i)$ is over the random perturbations described in Section 4.2, Equation (4). Equipped with this definition, we can take empirical expectations over $\tilde{q}_t \in \Delta(\mathcal{G} \times \mathcal{H})$ in the usual way. Observe that, in Algorithm 2, Equation (7), the optimization problem at round $t$, is equivalent to:

$$a_t \in \arg\min_{a \in \mathcal{A}} \max_{z \in \mathcal{Z}} \mathbb{E}_{j \sim q_t} \left[ r_t^j(a, z) \right].$$

Because the $d$ objectives in our reduction (Section B.2) correspond to each group-hypothesis pair $(g, h) \in \mathcal{G} \times \mathcal{H}$, $\mathcal{A}_t$ corresponds to $\Delta(\mathcal{Y})$, and $\mathcal{Z}_t$ always corresponds to $[0, 1]$, we can equivalently consider the min-max optimization problem at round $t \in [T]$:

$$p_t \in \arg\min_{p \in \Delta(\mathcal{Y})} \max_{\gamma \in [0,1]} \mathbb{E}_{(\tilde{g}, \tilde{h}) \sim \tilde{q}_t} \left[ r_t^{(\tilde{g}, \tilde{h})}(p, \gamma) \right], \tag{15}$$

where $r_t^{(\tilde{g}, \tilde{h})}$ is defined in Equation 9. The next lemma relates the optimization procedure of the $\mathcal{H}$-player in Algorithm 1 to the min-max optimization procedure of Equation (7) in Algorithm 2.

**Lemma B.3** (The optimization of the $\mathcal{H}$-player). *For any round $t \in [T]$, let $\{(\tilde{g}_i, \tilde{h}_i)\}_{i=1}^{M}$ denote the $M$ samples obtained by calling $\mathrm{OPT}_{(\mathcal{G},\mathcal{H})}^{\alpha}$ $M$ times in Algorithm 1, and denote $\tilde{q}_t \in \Delta(\mathcal{G} \times \mathcal{H})$ denote the corresponding empirical distribution (Definition B.4). Then, $p_t \in \Delta(\mathcal{Y})$ defined in Equation (15) above is equivalent to the distribution $(p, 1 - p) \in \Delta(\{-1, 1\})$ obtained from solving the linear program of the $\mathcal{H}$-player in Algorithm 1.*

*Proof.* Consider any round $t \in [T]$. Observe that Line 8 of Algorithm 1 is equivalent to solving the linear program for $\lambda \in \mathbb{R}$ and $\boldsymbol{p} := (p, 1 - p) \in \mathbb{R}^2$ :

$$\min \ \lambda$$
$$\text{s.t.} \ \sum \boldsymbol{p}_i = 1$$
$$\boldsymbol{p}^\top \tilde{L} e_i \leq \lambda \quad \forall i \in [2]$$
$$\boldsymbol{p}_i \geq 0 \quad \forall i \in [2]$$

where the payoff matrix $\tilde{L} \in [-1, 1]^{2 \times 2}$ has the coordinates $(y', y)$:

$$\tilde{L}_{(y', y)} := \frac{1}{M} \sum_{i=1}^{M} \tilde{\ell}_{x_t}((\tilde{g}_t^{(i)}, \tilde{h}_t^{(i)}), (y', y)),$$

and $e_i$ is the $i$th coordinate vector of $\mathbb{R}^2$. Let $\Delta(\mathcal{Y})$ denote the space of probability distributions over $\mathcal{Y}$. Let $\boldsymbol{p} = (p, 1 - p)$ and $\boldsymbol{z} = (\gamma, 1 - \gamma)$, and, as shorthand, denote $\boldsymbol{p}(-1) = 1 - p$, $\boldsymbol{p}(1) = p$, $\boldsymbol{z}(-1) = 1 - \gamma$, and $\boldsymbol{z}(1) = \gamma$.

By the equivalence of linear programs (LPs) to zero-sum min-max games (see, e.g., [BT97; Haz22]), obtaining the optimal $\boldsymbol{p}$ for this LP is the equivalent to solving:

$$
\begin{aligned}
\min_{\boldsymbol{p}\in\Delta(\mathcal{Y})} \max_{\boldsymbol{z}\in\Delta(\mathcal{Y})} \boldsymbol{p}^\top \tilde{L} \boldsymbol{z} &= \min_{\boldsymbol{p}\in\Delta(\mathcal{Y})} \max_{\boldsymbol{z}\in\Delta(\mathcal{Y})} \frac{1}{M} \sum_{y\in\mathcal{Y}} \sum_{y'\in\mathcal{Y}} \sum_{i=1}^{M} \boldsymbol{p}(y')\boldsymbol{z}(y)\tilde{\ell}_{x_t}((\tilde{g}_t^{(i)},\tilde{h}_t^{(i)}),(y',y)) \\
&= \min_{\boldsymbol{p}\in\Delta(\mathcal{Y})} \max_{\boldsymbol{z}\in\Delta(\mathcal{Y})} \sum_{y\in\mathcal{Y}} \sum_{y'\in\mathcal{Y}} \boldsymbol{p}(y')\boldsymbol{z}(y)\mathbb{E}_{(\tilde{g},\tilde{h})\sim\tilde{q}_t}\left[\tilde{\ell}_{x_t}((\tilde{g},\tilde{h}),(y',y))\right] \\
&= \min_{\boldsymbol{p}\in\Delta(\mathcal{Y})} \max_{\boldsymbol{z}\in\Delta(\mathcal{Y})} \mathbb{E}_{y'\sim\boldsymbol{p}}\mathbb{E}_{y\sim\mathrm{Ber}(\gamma)}\left[\mathbb{E}_{(\tilde{g},\tilde{h})\sim\tilde{q}_t}\left[\tilde{\ell}_{x_t}((\tilde{g},\tilde{h}),(y',y))\right]\right] \\
&= \min_{p\in\Delta(\mathcal{Y})} \max_{\gamma\in[0,1]} \mathbb{E}_{y'\sim\mathrm{Ber}(p)}\mathbb{E}_{y\sim\mathrm{Ber}(\gamma)}\left[\mathbb{E}_{(\tilde{g},\tilde{h})\sim\tilde{q}_t}\left[\tilde{\ell}_{x_t}((\tilde{g},\tilde{h}),(y',y))\right]\right] \\
&= \min_{p\in\Delta(\mathcal{Y})} \max_{\gamma\in[0,1]} \mathbb{E}_{(\tilde{g},\tilde{h})\sim\tilde{q}_t}\left[r_t^{(\tilde{g},\tilde{h})}(p,\gamma)\right]
\end{aligned}
$$

Above, the first equality just comes from definition of $\tilde{L}$, the second equality is from the Definition B.4 of the empirical distribution $\tilde{q}_t$, the third and fourth equalities are from the definition of $\boldsymbol{p}$ and $\boldsymbol{z}$ in the previous paragraph. The final equality is just from interchanging the order of expectation and the definition of $r_t^{(\tilde{g},\tilde{h})}(p,\gamma)$ in Equation (9). By this chain of inequalities, we see that obtaining the optimal $\boldsymbol{p}$ for the original LP corresponds exactly to the choice of $p_t$ in Equation (15). $\square$

Lemma B.3 tells us that the strategy of the $\mathcal{H}$-player (i.e., Lines 8 and 9 in Algorithm 1) to obtain $h_t$ is exactly the same as obtaining the minimizing $p_t$ in Equation (15). From the exposition above, this corresponds to the min-max optimization problem in Equation (7) when $q_t$ is $\tilde{q}_t$, the distribution over $\mathcal{G} \times \mathcal{H}$. We now proceed to prove a more general "meta-theorem" from which the regret guarantee of Algorithm 1 will follow once we plug in a specific FTPL algorithm for the $(\mathcal{G},\mathcal{H})$-player.

### B.5 Meta-algorithm for Online Multi-group Learning

We now present a meta-algorithm, Algorithm 4, and its corresponding Theorem B.4, the "meta-theorem" for online multi-group learning from which Theorem 4.1 follows. This is more general, however, than the guarantee in Theorem 4.1, and we emphasize that, through changing the no-regret algorithm for the $(\mathcal{G},\mathcal{H})$-player, we can obtain regret guarantees for settings other than the smoothed online learning setting of Theorem 4.1. Section 5 gives a couple of examples, which we elaborate in Appendix C.

Note that approximating the distribution of the $(\mathcal{G},\mathcal{H})$-player is crucial to obtain computational efficiency, as we cannot hope to enumerate $\mathcal{G}$ and $\mathcal{H}$ by explicitly representing $q_t$ in Algorithm 4. Thus, sampling $M$ times is a crucial "sparsification" step that allows us to implicitly access the distribution over $\mathcal{G} \times \mathcal{H}$ that the $(\mathcal{G},\mathcal{H})$-player maintains.

We prove Theorem B.4 through techniques similar to the regret guarantee proof of Algorithm 2 in [Lee+22]. Namely, we observe that by using the reduction outlined in Section B.2 with the $\mathcal{G} \times \mathcal{H}$-dimensional loss

$$
\begin{aligned}
r_t^{(\tilde{g},\tilde{h})}(p,\gamma) &= \mathbb{E}_{h(x_t)\sim p}\mathbb{E}_{y\sim\mathrm{Ber}(\gamma)}\left[\tilde{g}(x_t)\left(\ell(h(x_t),y)-\ell(\tilde{h}(x_t),y)\right)\right] \\
&= \mathbb{E}_{h(x_t)\sim p}\mathbb{E}_{y\sim\mathrm{Ber}(\gamma)}\left[\tilde{\ell}\left((\tilde{g},\tilde{h}),(h(x_t),y)\right)\right]
\end{aligned}
$$

we may obtain a bound on the multi-group regret of our Algorithm 4 by obtaining an adversary-moves-first regret guarantee (Definition B.2). For simplicity, we prove this for the case of binary actions $\mathcal{Y} = \{-1,1\}$; we outline how to obtain a similar theorem for multi-class action spaces in Appendix C.

**Theorem B.4** (Meta-Theorem for Online Multi-group Learning). *Let $\mathcal{X}$ be a context space, let $\mathcal{Y} := \{-1,1\}$ be a binary action space, and let $\mathcal{H}$ be a hypothesis class of functions $h : \mathcal{X} \to \mathcal{Y}$, and let $\mathcal{G}$ be a collection of groups, $g : \mathcal{X} \to \{0,1\}$. Let $\ell : \mathcal{Y} \times \mathcal{Y} \to [0,1]$ be a bounded loss function. Suppose the $(\mathcal{G},\mathcal{H})$-player of Algorithm 4 is instantiated with a no-regret (maximization) algorithm operating over $\mathcal{H} \times \mathcal{G}$ that has the guarantee that for any sequence of losses of length $T$ bounded in $[-1,1]$, by playing (a possibly implicit distribution) $q_t$, it has regret at most $R(T)$ in expectation.*

**Algorithm 4** Meta-algorithm for Online Multi-group Learning ($\mathcal{Y} = \{-1, 1\}$)

**Input:** $M \in \mathbb{N}$, the number of samples to take from $q_t$ at each step.

1: **for** $t = 1, 2, 3, \ldots, T$ **do**
2:    Receive a context $x_t$ from Nature.
3:    For any $y', y \in \mathcal{Y}$, construct the loss:

$$\tilde{\ell}_{x_t}((\tilde{g}, \tilde{h}), (y', y)) := \tilde{g}(x_t)\left(\ell(y', y) - \ell(\tilde{h}(x_t), y)\right).$$

4:    $(\mathcal{G}, \mathcal{H})$**-player:** With access to the entire history of the past $t - 1$ rounds and $x_t$, with access to $\mathrm{OPT}^{\alpha}_{(\mathcal{G}, \mathcal{H})}$, run a no-regret algorithm over the benchmark class $\mathcal{G} \times \mathcal{H}$ to output a distribution $q_t \in \Delta(\mathcal{G} \times \mathcal{H})$.
5:    Construct an empirical approximation $\tilde{q}_t$ of $q_t$ by sampling from $q_t$ $M$ times, obtaining a collection of pairs $\{(\tilde{h}_t^{(i)}, \tilde{g}_t^{(i)})\}_{i=1}^{M}$ from $\mathcal{G} \times \mathcal{H}$.
6:    $\mathcal{H}$**-player:** Call $\mathrm{OPT}_{\mathcal{H}}$ twice on the singleton datasets $\{(x_t, -1)\}$ and $\{(x_t, 1)\}$ with the 0-1 loss, obtaining:

$$h'_1 \in \operatorname*{arg\,min}_{h^* \in \mathcal{H}} \mathbf{1}\left\{h^*(x_t) \neq 1\right\}, \quad h'_{-1} \in \operatorname*{arg\,min}_{h^* \in \mathcal{H}} \mathbf{1}\left\{h^*(x_t) \neq -1\right\}.$$

7:    $\mathcal{H}$**-player:** Solve the linear program

$$\min_{p, \lambda \in \mathbb{R}} \quad \lambda$$

$$\text{subj. to} \quad \sum_{i=1}^{M} p\tilde{\ell}_{x_t}((\tilde{g}_t^{(i)}, \tilde{h}_t^{(i)}), (h'_1(x_t), y)) + (1 - p)\tilde{\ell}_{x_t}((\tilde{g}_t^{(i)}, \tilde{h}_t^{(i)}), (h'_{-1}(x_t), y)) \leq \lambda$$

$$\forall y \in \{-1, 1\}$$

$$0 \leq p \leq 1.$$

8:    Sample $b \sim \mathrm{Ber}(p)$, where $b \in \{-1, 1\}$, let $h_t = h'_b$.
9:    Learner commits to the action $\hat{y}_t = h_t(x_t)$; Nature reveals $y_t$.
10:    Learner incurs the loss $\ell(\hat{y}_t, y_t)$.
11:    The $(\mathcal{G}, \mathcal{H})$-player draws $(\tilde{g}, \tilde{h}) \sim \tilde{q}_t$ and incurs the loss $\tilde{\ell}_{x_t}((\tilde{g}, \tilde{h}), (\hat{y}_t, y_t))$.
12: **end for**

---

*Then, with expectation over any randomness of the $(\mathcal{G}, \mathcal{H})$-player and Nature, Algorithm 1 obtains the multi-group regret guarantee of:*

$$\mathbb{E}[\mathrm{Reg}_T(\mathcal{H}, g)] \leq \sum_{t=1}^{T} v_t^A + \sum_{t=1}^{T} \mathbb{E}[\epsilon_t(M)] + R(T), \tag{16}$$

*for all $g \in \mathcal{G}$, where*

$$v_t^A := \max_{\gamma \in [0,1]} \min_{p_t \in \Delta(\mathcal{H})} \max_{(\tilde{g}, \tilde{h}) \in \mathcal{G} \times \mathcal{H}} \mathbb{E}_{h \sim p_t, y \sim \mathrm{Ber}(\gamma)}\left[\tilde{g}(x_t)(\ell(h(x_t), y) - \ell(\tilde{h}(x_t), y))\right] \tag{17}$$

*and $\epsilon_t(M)$ is the error incurred from estimating $q_t$ with $\tilde{q}_t$ with $M$ samples at step $t \in [T]$.*

*Proof.* From the perspective of the $(\mathcal{G}, \mathcal{H})$-player, who is running a regret maximization algorithm, the following game is being played. For $t = 1, 2, 3, \ldots, T$:

- Receive a context $x_t \in \mathcal{X}$ from Nature, possibly adversarially and depending on the past $t - 1$ rounds.
- Play a pair $(\tilde{h}_t, \tilde{g}_t) \in \mathcal{H} \times \mathcal{G}$, possibly randomly and dependent on the last $t - 1$ rounds, where pairs are functions

$$(h, g) : \mathcal{X} \to \mathcal{Y} \times \{0, 1\}.$$

- Commit to the action $\tilde{y}_t := (\tilde{h}_t(x_t), \tilde{g}_t(x_t)) \in \mathcal{Y} \times \{0, 1\}$.

- An adversary (the $\mathcal{H}$-player's prediction *and* Nature) reveals $(\hat{y}_t, y_t) \in \mathcal{Y} \times \mathcal{Y}$, and we incur the loss:

$$\tilde{\ell}((\tilde{g}, \tilde{h}), (\hat{y}_t, y_t)) := \tilde{g}(x_t)\left(\ell(\hat{y}_t, y_t) - \ell(\tilde{h}_t(x_t), y_t)\right).$$

  Here, $\hat{y}_t = h_t(x_t)$, which is a random variable depending on sampling from $p_t$, the Bernoulli distribution of the $\mathcal{H}$-player at round $t$.

Note that the $(\mathcal{G}, \mathcal{H})$-player is attempting to *maximize* this loss.

Let $h_1, \ldots, h_T$ be the sequence of hypotheses chosen by the $\mathcal{H}$-player, and let $y_1, \ldots, y_T$ be the sequence of adversarially chosen outcomes. Then, by the regret guarantee of the $(\mathcal{G}, \mathcal{H})$-player's no-regret algorithm algorithm, for any $(h^*, g^*) \in \mathcal{H} \times \mathcal{G}$:

$$\sum_{t=1}^{T} \mathbb{E}\left[\tilde{\ell}((g^*, h^*), (h_t(x_t), y_t))\right] - \sum_{t=1}^{T} \mathbb{E}_{(\tilde{g}_t, \tilde{h}_t)\sim q_t}\left[\tilde{\ell}((\tilde{g}_t, \tilde{h}_t), (h_t(x_t), y_t))\right] \leq R(T),$$

where the expectation is over the distributions $q_t$ over $\Delta(\mathcal{H} \times \mathcal{G})$ that the $(\mathcal{G}, \mathcal{H})$-player commits to at each round *and* the random choices of Nature and the $\mathcal{H}$-player's random choice of $\hat{y}_t$ at each round. To ease notation, we keep the subscript in the expectation over $h_t$ hidden, noting that $h_t(x_t) \sim p_t$ is a random variable throughout. For instance, in the case of Theorem 4.1, this is a random process determined by the $(\mathcal{G}, \mathcal{H})$-player sampling $n$ perturbation terms and calling the $(\mathcal{G}, \mathcal{H})$-oracle to obtain the random variable $(\tilde{g}_t, \tilde{h}_t)$. Instantiating the regret bound for all $(h^*, g^*)$ gives us:

$$\max_{(h^*, g^*)\in\mathcal{H}\times\mathcal{G}} \sum_{t=1}^{T} \mathbb{E}\left[\tilde{\ell}((g^*, h^*), (h_t(x_t), y_t))\right] \leq \sum_{t=1}^{T} \mathbb{E}_{(\tilde{g}_t, \tilde{h}_t)\sim q_t}\left[\tilde{\ell}((\tilde{g}_t, \tilde{h}_t), (h_t(x_t), y_t))\right] + R(T).$$

However, we do not have direct access to $q_t$, the implicit distribution over $\mathcal{G} \times \mathcal{H}$, but we have an approximation $\tilde{q}_t$. In Algorithm 2, $(\tilde{g}_t, \tilde{h}_t)$ is drawn according to $\tilde{q}_t$, the empirical distribution over $\Delta(\mathcal{H} \times \mathcal{G})$ formed by drawing $M$ samples $\{(\tilde{h}_t^{(i)}, \tilde{g}_t^{(i)})\}_{i=1}^{M}$ from $\tilde{q}_t$. Fixing $x_t, y_t$, and $h_t(x_t)$, let $\epsilon_t(M)$ be the error we incur from replacing the true distribution $q_t$ by the empirical distribution $\tilde{q}_t$, which we can handle through uniform convergence on the samples $\{(\tilde{h}_t^{(i)}, \tilde{g}_t^{(i)})\}_{i=1}^{M}$ (see Lemma B.6):

$$\left|\mathbb{E}_{(\tilde{g}, \tilde{h})\sim\tilde{q}_t}[\tilde{\ell}((\tilde{g}, \tilde{h}), (h_t(x_t), y_t))] - \mathbb{E}_{(g, h)\sim q_t}[\tilde{\ell}((g, h), (h_t(x_t), y_t))]\right| \leq \epsilon_t(M).$$

Adding the estimation error at each $t \in [T]$, our regret bound becomes:

$$\max_{(h^*, g^*)\in\mathcal{H}\times\mathcal{G}} \sum_{t=1}^{T} \mathbb{E}\left[\tilde{\ell}((g^*, h^*), (h_t(x_t), y_t))\right] \leq \sum_{t=1}^{T} \mathbb{E}_{(\tilde{g}_t, \tilde{h}_t)\sim q_t}\left[\tilde{\ell}((\tilde{g}_t, \tilde{h}_t), (h_t(x_t), y_t))\right] + R(T)$$

$$\leq \sum_{t=1}^{T} \mathbb{E}_{(\tilde{g}_t, \tilde{h}_t)\sim\tilde{q}_t}\left[\tilde{\ell}((\tilde{g}_t, \tilde{h}_t), (h_t(x_t), y_t))\right] + \sum_{t=1}^{T} \epsilon_t(M) + R(T).$$

Now, we aim to bound the terms $\mathbb{E}_{(\tilde{g}_t, \tilde{h}_t)\sim\tilde{q}_t}\left[\tilde{\ell}((\tilde{g}_t, \tilde{h}_t), (h_t(x_t), y_t))\right]$. At round $t \in [T]$, Algorithm 4 chooses the best response $h_t \in \mathcal{H}$ by solving a linear program for a Bernoulli parameter $p$ and then sampling $h$ from $\mathrm{Ber}(p)$. This is equivalent to sampling $h_t(x_t) \sim p_t$, where $p_t := (p, 1-p)$ on $\Delta(\mathcal{Y})$. We use sampling from this Bernoulli distribution and sampling from $p_t$ interchangeably. By Lemma B.3, this is equivalent to solving the min-max optimization problem in Equation (15)

$$p_t \in \arg\min_{p\in\Delta(\mathcal{Y})} \max_{\gamma\in[0,1]} \mathbb{E}_{(\tilde{g}_t, \tilde{h}_t)\sim\tilde{q}_t}\left[r_t^{(\tilde{g}_t, \tilde{h}_t)}(p, \gamma)\right],$$

which, by definition of $r_t^{(\tilde{g}, \tilde{h})}(p, \gamma)$, is equivalent to:

$$p_t \in \arg\min_{p_t\in\Delta(\mathcal{Y})} \max_{\gamma\in[0,1]} \mathbb{E}_{(\tilde{g}_t, \tilde{h}_t)\sim\tilde{q}_t}\left[\mathbb{E}_{h_t(x_t)\sim p_t}\mathbb{E}_{y\sim\mathrm{Ber}(\gamma)}\left[\tilde{g}_t(x_t)\left(\ell(h_t(x_t), y) - \ell(\tilde{h}_t(x_t), y)\right)\right]\right].$$
$$(18)$$

The inner expectations are linear in $p_t$ and $\gamma$, and taking the outer expectation over $\tilde{q}_t$ maintains linearity. Therefore, this is a convex and concave min-max optimization problem, and von Neumann's

minimax theorem [NMR44] applies, allowing us to swap the order of minimization and maximization. Therefore

$$\min_{p_t \in \Delta(\mathcal{Y})} \max_{\gamma \in [0,1]} \mathbb{E}_{(\tilde{g}_t, \tilde{h}_t) \sim \tilde{q}_t} \left[ \mathbb{E}_{h_t(x_t) \sim p_t} \mathbb{E}_{y \sim \mathrm{Ber}(\gamma)} \left[ \tilde{g}_t(x_t) \left( \ell(h_t(x_t), y) - \ell(\tilde{h}_t(x_t), y) \right) \right] \right]$$

$$= \max_{\gamma \in [0,1]} \min_{p_t \in \Delta(\mathcal{Y})} \mathbb{E}_{(\tilde{g}_t, \tilde{h}_t) \sim \tilde{q}_t} \left[ \mathbb{E}_{h_t(x_t) \sim p_t} \mathbb{E}_{y \sim \mathrm{Ber}(\gamma)} \left[ \tilde{g}_t(x_t) \left( \ell(h_t(x_t), y) - \ell(\tilde{h}_t(x_t), y) \right) \right] \right]$$

$$\leq \max_{\gamma \in [0,1]} \min_{p_t \in \Delta(\mathcal{Y})} \max_{(\tilde{g}, \tilde{h}) \in \mathcal{G} \times \mathcal{H}} \mathbb{E}_{h_t(x_t) \sim p_t, y \sim \mathrm{Ber}(\gamma)} \left[ \tilde{g}(x_t) \left( \ell(h_t(x_t), y) - \ell(\tilde{h}(x_t), y) \right) \right] = v_t^A$$

where the first equality is from the minimax theorem and the second inequality is just because averages are less than or equal to maximums. Combining all the inequalities, we obtain

$$\max_{(h^*, g^*) \in \mathcal{H} \times \mathcal{G}} \sum_{t=1}^{T} \tilde{\ell}((g^*, h^*), (h_t(x_t), y_t)) \leq \sum_{t=1}^{T} v_t^A + \sum_{t=1}^{T} \epsilon_t(M) + R(T),$$

and our theorem follows from taking an expectation on both sides and substituting back the definition of

$$\tilde{\ell}((g^*, h^*), (h(x_t), y_t)) := g^*(x_t) \left( \ell(h(x_t), y_t) - \ell(h^*(x_t), y_t) \right),$$

because each $\tilde{\ell}((g^*, h^*), (h(x_t), y_t))$ is simply the per-round regret. $\qquad \square$

With Theorem B.4 in hand, it remains to make sure that the two terms $\sum_{t=1}^{T} v_t^A$ and $\sum_{t=1}^{T} \mathbb{E}[\epsilon_t(M)]$ are both $o(T)$. Then, if we have a no-regret algorithm with $o(T)$ regret while only accessing $\mathcal{G}$ and $\mathcal{H}$ through the $\mathrm{OPT}_{(\mathcal{G}, \mathcal{H})}^\alpha$ oracle, we will have a multi-group online learning algorithm. The next two lemmas show that both terms are, indeed, $o(T)$.

First, we bound the values $v_t^A$, which are known as the "AMF values" in the framework of [Lee+22].

**Lemma B.5.** *For any $t \in [T]$, the AMF value of the game at round $t$, is nonpositive, i.e.*

$$v_t^A := \max_{\gamma \in [0,1]} \min_{p_t \in \Delta(\mathcal{H})} \max_{(h,g) \in \mathcal{H} \times \mathcal{G}} \mathbb{E}[g(x_t)(\ell(h_t(x_t), y) - \ell(h(x_t), y))] \leq 0,$$

*where the expectation is taken over $h_t \sim p_t$ and $y \sim \mathrm{Ber}(\gamma)$.*

*Proof.* Fix any parameter $\gamma \in [0, 1]$ for the max player. Then, for any $(h, g) \in \mathcal{H} \times \mathcal{G}$,

$$\mathbb{E}\left[g(x_t)(\ell(h_t(x_t), y) - \ell(h(x_t), y))\right] = \gamma \mathbb{E}_{h_t \sim p_t}[g(x_t)(\ell(h_t(x_t), 1) - \ell(h(x_t), 1))]$$
$$+ (1 - \gamma) \mathbb{E}_{h_t \sim p_t}[g(x_t)(\ell(h_t(x_t), -1) - \ell(h(x_t), -1))].$$

Expanding the expectation over $h_t \sim p_t$, this is equivalent to:

$$g(x_t) \sum_{h' \in \mathcal{H}} p_{h'}^t (\gamma \ell(h'(x_t), 1) + (1 - \gamma)\ell(h'(x_t), 0)) - g(x_t)(\gamma \ell(h(x_t), 1) + (1 - \gamma)\ell(h(x_t), 0)),$$

so it suffices to find $p_t \in \Delta(\mathcal{H})$ such that, for all $(h, g) \in \mathcal{H} \times \mathcal{G}$,

$$g(x_t) \sum_{h' \in \mathcal{H}} p_{h'}^t (\gamma \ell(h'(x_t), 1) + (1 - \gamma)\ell(h'(x_t), 0)) \leq g(x_t)(\gamma \ell(h(x_t), 1) + (1 - \gamma)\ell(h(x_t), 0)).$$

The $g(x_t)$ value is the same for both sides, so it really suffices to find the $p_t \in \Delta(\mathcal{H})$ such that, for all $h \in \mathcal{H}$,

$$p_{h'}^t (\gamma \ell(h'(x_t), 1) + (1 - \gamma)\ell(h'(x_t), 0)) \leq \gamma \ell(h(x_t), 1) + (1 - \gamma)\ell(h(x_t), 0).$$

Because we know $\gamma$, the Bernoulli parameter for the true distribution of $y \mid x_t$, we can choose $p^t \in \Delta(\mathcal{H})$ to put all its mass on the $h^* \in \mathcal{H}$ that minimizes this risk, i.e.

$$h^* \in \arg\min_{h \in \mathcal{H}} \mathbb{E}_y[\ell(h(x_t), y) \mid x_t].$$

This is, by definition, exactly the $h^*$ such that, for any $h \in \mathcal{H}$,

$$\gamma \ell(h^*(x_t), 1) + (1 - \gamma)\ell(h^*(x_t), -1) \leq \gamma \ell(h(x_t), 1) + (1 - \gamma)\ell(h(x_t), -1).$$

Therefore, we have that:

$$
\begin{aligned}
v_t^A &= \max_{\gamma \in [0,1]} \min_{p_t \in \Delta(\mathcal{H})} \max_{(h,g) \in \mathcal{H} \times \mathcal{G}} \mathbb{E}_{h' \sim p_t, y \sim \text{Ber}(\gamma)}[g(x_t)(\ell(h'(x_t), y) - \ell(h(x_t), y))] \\
&\leq \max_{\gamma \in [0,1], (h,g) \in \mathcal{H} \times \mathcal{G}} \mathbb{E}_{y \sim \text{Ber}(\gamma)}[g(x_t)(\ell(h^*(x_t), y) - \ell(h(x_t), y))] \\
&\leq 0.
\end{aligned}
$$

The final inequality follows from our argument above. $\qquad\square$

Next, we bound the expected approximation error we incur from replacing $q_t$ with the empirical distribution $\tilde{q}_t$ obtained from the $M$ samples $\{(\tilde{g}_t^{(i)}, \tilde{h}_t^{(i)})\}_{i=1}^M$, which we denoted as $\epsilon_t(M)$. This comes from a standard uniform convergence argument.

**Lemma B.6.** *Let $t \in [T]$ and $x_t \in \mathcal{X}$ be fixed, and consider the function $\tilde{\ell}_{x_t}((g,h),(y',y)) := g(x_t)(\ell(y',y) - \ell(h(x_t), y))$. Let $|\mathcal{Y}| = k < \infty$. If $M \geq T^{1+\delta}$, where $\delta = \Omega(\frac{\log(\log T + \log k)}{\log T})$, then over the randomness of drawing $M$ samples $\{(\tilde{g}_t^{(i)}, \tilde{h}_t^{(i)})\}_{i=1}^M$ to construct the empirical distribution $\tilde{q}_t$ described in Definition B.4, for all $y', y \in \mathcal{Y}$, let $\epsilon_t(M)$ be defined as the supremum*

$$
\epsilon_t(M) := \sup_{(y',y) \in \mathcal{Y} \times \mathcal{Y}} \left| \mathbb{E}_{(\tilde{g},\tilde{h}) \sim \tilde{q}_t}[\tilde{\ell}_{x_t}((\tilde{g}, \tilde{h}), (y',y))] - \mathbb{E}_{(g,h) \sim q_t}[\tilde{\ell}_{x_t}((g,h),(y',y))] \right|,
$$

*and, over all $T$ rounds,*

$$
\mathbb{E}\left[\sum_{i=1}^T \epsilon_t(M)\right] \leq 2\sqrt{T}.
$$

*Proof.* Fix any round $t \in [T]$. We use a standard uniform convergence argument to ensure that the empirical distribution $\tilde{q}_t$ and the true distribution $q_t$ are close for the function $\tilde{\ell}_{x_t}((g,h),(y',y))$ for all $y', y \in \mathcal{Y}$. We assume that $\mathcal{Y}$ is finite, and denote $k := |\mathcal{Y}|$.

Let $\{(\tilde{g}_t^{(i)}, \tilde{h}_t^{(i)})\}_{i=1}^M$ denote the $M$ random samples from $\mathcal{G} \times \mathcal{H}$. We note that the expectation over $\tilde{q}_t$ is the same as:

$$
\mathbb{E}_{(\tilde{g},\tilde{h}) \sim \tilde{q}_t}\left[\tilde{\ell}((\tilde{g}, \tilde{h}), (y',y))\right] = \frac{1}{M} \sum_{i=1}^M \tilde{\ell}\left((\tilde{g}_t^{(i)}, \tilde{h}_t^{(i)}), (y',y)\right).
$$

Consider the empirical process

$$
\sup_{(y',y) \in \mathcal{Y}^2} \left| \frac{1}{M} \sum_{i=1}^M \underbrace{\tilde{\ell}((\tilde{g}_t^{(i)}, \tilde{h}_t^{(i)}), (y',y)) - \mathbb{E}_{(g,h) \sim q_t}\left[\tilde{\ell}((g,h),(y',y))\right]}_{Z_i(y',y)} \right|.
$$

For notational simplicity, let us refer to this empirical process as:

$$
\sup_{(y',y) \in \mathcal{Y}^2} \left| \frac{1}{M} \sum_{i=1}^M Z_i(y',y) \right| = \epsilon_t(M).
$$

Because $\ell(\cdot, \cdot) \in [0,1]$, we know $\tilde{\ell} \in [-1,1]$. Moreover, $|\mathcal{Y}^2| = k^2$. We can now use Hoeffding's inequality and a union bound to obtain:

$$
\mathbb{P}\left[\sup_{(y',y) \in \mathcal{Y}^2} \left| \frac{1}{M} \sum_{i=1}^M Z_i(y',y) \right| \geq \varepsilon\right] \leq k^2 \exp(-2M\varepsilon^2) \leq k^2 \exp(-2T^{1+\delta}\varepsilon^2).
$$

By the elementary integral inequality $\mathbb{E}[X] \leq \int_0^\infty \mathbb{P}[X \geq t]dt$, we obtain:

$$\mathbb{E}\left[\sup_{(y',y)\in\mathcal{Y}^2}\left|\frac{1}{M}\sum_{i=1}^M Z_i(y',y)\right|\right] \leq \int_0^\infty \mathbb{P}\left[\sup_{(y',y)\in\mathcal{Y}^2}\left|\frac{1}{M}\sum_{i=1}^M Z_i(y',y)\right| \geq t\right]dt$$

$$\leq \int_0^w \mathbb{P}\left[\sup_{(y',y)\in\mathcal{Y}^2}\left|\frac{1}{M}\sum_{i=1}^M Z_i(y',y)\right| \geq t\right]dt + \int_w^\infty k^2\exp(-2T^{1+\delta}t^2)dt$$

$$\leq w + \int_w^\infty k^2\exp(-2T^{1+\delta}t^2)dt$$

$$\leq w + k^2\exp(-2T^{1+\delta}w^2),$$

where $w > 0$ is an arbitrary parameter. Set $w = \sqrt{\frac{1}{T}}$. If $\delta \geq \frac{\log\left(2\log(k)+\frac{1}{2}\log(T)\right)}{2\log T}$, then:

$$\mathbb{E}\left[\sup_{(y',y)\in\mathcal{Y}^2}\left|\frac{1}{M}\sum_{i=1}^M Z_i(y',y)\right|\right] = \mathbb{E}[\epsilon_t(M)] \leq 2\sqrt{\frac{1}{T}}.$$

Therefore, summing up over all $T$ rounds, we obtain $\sum_{t=1}^T \mathbb{E}[\epsilon_t(M)] \leq 2\sqrt{T}$. $\qquad\square$

## B.6  Instantiating the meta-algorithm for Theorem 4.1

Finally, we instantiate Theorem B.4 with a concrete no-regret algorithm for the $(\mathcal{G},\mathcal{H})$-player to obtain Theorem 4.1. We employ the specific no-regret algorithm of [Blo+22] for our $(\mathcal{G},\mathcal{H})$-player, restated here for reference.

**Theorem B.7** (Smoothed FTPL of [Blo+22]). *Let $\mathcal{F} : \mathcal{X} \to [-1,1]$ be a function class and let $\ell$ be a loss function that is $L$-Lipscchitz in both arguments. Suppose further that we are in the* smoothed *online learning setting (see Section 4.1) where each $x_i$ are drawn from a distribution that is $\sigma$-smooth with respect ot some base measure $\mathcal{B}$ on $\mathcal{X}$. Let*

$$\pi_{t,n}(f) := \sum_{i=1}^n \frac{f(z_{t,i})\gamma_{t,i}}{\sqrt{n}},$$

*where $z_{t,i} \sim \mathcal{B}$ are independent and the $\gamma_{t,i}$ are independent standard Gaussian variables. Suppose that $\alpha \geq 0$ and consider the algorithm which uses an $\alpha$-approximate oracle for $\mathcal{F}$ (see Definition 2.1) to choose $f_t$ according to*

$$\sum_{s=1}^{t-1}\ell(f_t(x_s),y_s) + \eta\pi_{t,n}(f_t) \leq \inf_{f^*\in\mathcal{F}}\sum_{s=1}^{t-1}\ell(f^*(x_s),y_s) + \pi_{t,n}(f^*) + \alpha,$$

*and let $\hat{y}_t = f_t(x_t)$. If $\mathcal{F}$ and $y_t$ are binary valued, with the VC dimension of $\mathcal{F}$ bounded by $d \geq 1$, then for $n = T/\sqrt{\sigma}$ and $\eta = \sqrt{\frac{T\log(TL/\sigma)}{\sigma}}$*

$$\mathbb{E}[\mathrm{Reg}_T(f_t)] \leq C\left(\sqrt{\frac{dT\log T}{\sigma}} + \alpha T\right),$$

*where $C > 0$ is some absolute constant.*

This is precisely what the $(\mathcal{G},\mathcal{H})$-player does in Algorithm 1. Let $\mathcal{G} := \{g \subseteq \mathcal{X} : g \in \mathcal{G}\}$ be a collection of groups, represented as Boolean functions $g : \mathcal{X} \to \{0,1\}$. Let $\mathcal{H}$ be a hypothesis class of binary-valued functions $h : \mathcal{X} \to \{-1,1\}$. To be clear, the we map Theorem B.7 to our Algorithm 1 in the following way:

- Let $\mathcal{F}$ of Theorem B.7 be the class of functions in $[-1,1]^{\mathcal{X}}$ defined by:

$$\mathcal{F} := \{x \mapsto \tilde{g}(x)\tilde{h}(x) : \tilde{g} \in \mathcal{G}, \tilde{h} \in \mathcal{H}\}.$$

  Note that each $f \in \mathcal{F}$ maps to $\{-1,0,1\}$.

- Let the loss function in Theorem B.7 be the loss of the $(\mathcal{G}, \mathcal{H})$-player on $x \in \mathcal{X}$:

$$\tilde{\ell}((\tilde{g}, \tilde{h}), (h(x), y)) := \tilde{g}(x) \left( \ell(h(x), y) - \ell(\tilde{h}(x), y) \right).$$

In terms of $\mathcal{F}$ above, we can rewrite this as:

$$\tilde{\ell}(f(x), (y', y)) := \begin{cases} 0 & \text{if } f(x) = 0 \\ \ell(y', y) - \ell(-1, y) & \text{if } f(x) = -1 \\ \ell(y', y) - \ell(1, y) & \text{if } f(x) = 1. \end{cases}$$

This loss function has the signature $\tilde{\ell} : \{-1, 0, 1\} \times \{-1, 1\}^2 \to [-1, 1]$ because $\ell(\cdot, \cdot) \in [0, 1]$. It is also 2-Lipschitz in both arguments.

- It remains to make sure that ternary-valued functions taking values in $\{-1, 0, 1\}$ do not break the proof of Theorem B.7. In the proof of Theorem B.7 in [Blo+22], the binary-valued function case where $f$ has range $\{-1, 1\}$ is handled by embedding $\{-1, 1\}$ into the real line. There are two main parts of the proof that rely on this assumption that $f$ has range $\{-1, 1\}$ that easily maintain when $f$ has range $\{-1, 0, 1\}$.
  - First, in Lemma 34 of [Blo+22], the authors use $L$-Lipschitzness and the simple fact that $|f(x) - f'(x)| \leq (f(x) - f'(x))^2$ when $f \in \{-1, 1\}$. This still holds when $f \in \{-1, 0, 1\}$.
  - Second, [Blo+22] also use the fact that $\|f\|_{L_2} = 1$ for all $f \in \mathcal{F}$, which is also true for $f \in \{-1, 0, 1\}$.

Finally, the rest of the proof in Lemma 35 of [Blo+22] relies only on the Lipschitzneess of $\tilde{\ell}$ to employ standard smoothness arguments and Rademacher contraction, which we have already established.

- Putting all this together, the $(\mathcal{G}, \mathcal{H})$-player in Algorithm 1 essentially runs the algorithm of B.7, with the caveat that it calls the $\text{OPT}^\alpha_{(\mathcal{G}, \mathcal{H})}$ oracle $M$ times to get the empirical approximation $\tilde{q}_t$ of the true implicit distribution $q_t$ over $\mathcal{G} \times \mathcal{H}$. This implicit distribution is defined by the random process of drawing the $n$ perturbations and calling the optimization oracle.

Therefore, by Lemmas B.6, B.5, and Theorem B.7 applied to the "meta-theorem" Theorem B.4, we immediately obtain our main theorem, Theorem 4.1. We now restate Theorem 4.1 as Corollary B.7.1 with the specified choices of parameters.

**Corollary B.7.1** (Theorem 4.1, with parameters specified). *Let $\mathcal{Y} = \{-1, 1\}$ be a binary action space, $\mathcal{H} \subseteq \{-1, 1\}^{\mathcal{X}}$ be a binary-valued hypothesis class, $\mathcal{G} \subseteq 2^{\mathcal{X}}$ be a (possibly infinite) collection of groups, and $\ell : \{-1, 1\} \times \{-1, 1\} \to [0, 1]$ be a bounded loss function. Let the VC dimensions of $\mathcal{H}$ and $\mathcal{G}$ both be bounded by $d$. Let $\alpha \geq 0$ be the approximation error of the oracle $\text{OPT}^\alpha_{(\mathcal{G}, \mathcal{H})}$. If we are in the $\sigma$-smooth online learning setting, then, for $M \geq T^{1+\delta}, \delta \geq \frac{\log\left(2 \log(k) + \frac{1}{2} \log(T)\right)}{2 \log T}, n = T/\sqrt{\sigma}$, and $\eta = \sqrt{\frac{T \log(T/\sigma)}{\sigma}}$, Algorithm 1 achieves, for each $g \in \mathcal{G}$:*

$$\mathbb{E}[\text{Reg}_T(\mathcal{H}, g)] \leq \sqrt{\frac{dT \log T}{\sigma}} + \alpha T,$$

*where the expectation is over all the randomness of the $(\mathcal{G}, \mathcal{H})$-player's perturbations and the $\mathcal{H}$-player's Bernoulli choices.*

## C    Other instantiations of the meta-algorithm

It is be clear from the statement of Theorem B.4 that our "meta-algorithm" Algorithm 4 straightforwardly applies for other online learning settings as well, so long as we adopt an appropriate strategy for the $(\mathcal{G}, \mathcal{H})$-player. In this section, we give a few examples of this flexibility for discrete action spaces in the smoothed online setting and the

### C.1    Multi-class action spaces

Instead of $|\mathcal{Y}| = 2$, we can let $|\mathcal{Y}| = K$, more generally. In this case, a straightforward extension of Theorem B.7 allows us to embed $\mathcal{Y} \cup \{0, 1\}$ into the real line, and we generalize to considering

the Natarajan dimension [Nat89] of $\mathcal{H}$ instead of the VC dimension. Rademacher contraction and Lipschitzness still apply to $\tilde{\ell}$, so with just a difference in the absolute constant, we obtain the following multi-class analogue of Theorem 4.1 as a corollary. Thus, for the $(\mathcal{G}, \mathcal{H})$-player in Algorithm 1, we can just use the same exact algorithm outlined in Theorem B.7.

We make a small change to the $\mathcal{H}$-player in Algorithm 1. In the multi-class action space setting, we need to make $K$ calls to the $\mathrm{OPT}_{\mathcal{H}}$ oracle and solve a $K \times K$ size linear program for the $\mathcal{H}$-player at each step. For completeness, we present the algorithm for the $K$-class action spaces here, as Algorithm 5.

---

**Algorithm 5** Algorithm for Group Oracle Efficiency (multi-class)

---

**Input:** Perturbation strength $\eta > 0$; number of $\mathrm{OPT}^{\alpha}_{(\mathcal{G}, \mathcal{H})}$ calls $M \in \mathbb{N}$.

1: **for** $t = 1, 2, 3, \ldots, T$ **do**
2:      Receive a context $x_t \sim \mu_t$ from Nature.
3:      **for** $i = 1, 2, 3, \ldots, M$ **do**
4:          $(\mathcal{G}, \mathcal{H})$-**player:** Draw $n$ hallucinated examples as in Equation (4) to construct $\pi^{\mathrm{bin}}_{t,n}$.
5:          $(\mathcal{G}, \mathcal{H})$-**player:** Using the entire history $\{(\hat{y}_s, y_s)\}^{t-1}_{s=1}$ so far, call $\mathrm{OPT}^{\alpha}_{(\mathcal{G}, \mathcal{H})}$ to obtain $(\tilde{g}_t(i), \tilde{h}^{(i)}_t) \in \mathcal{G} \times \mathcal{H}$ satisfying:

$$\sum_{s=1}^{t-1} \tilde{\ell}_{x_s}((\tilde{g}, \tilde{h}), (\hat{y}_s, y_s)) + \pi^{\mathrm{bin}}_{t,n}(\tilde{g}, \tilde{h}, \eta)$$

$$\geq \sup_{(g^*, h^*) \in \mathcal{G} \times \mathcal{H}} \sum_{s=1}^{t-1} \tilde{\ell}_{x_s}((g^*, h^*), (\hat{y}_s, y_s)) + \pi^{\mathrm{bin}}_{t,n}(g^*, h^*, \eta) - \alpha \quad (19)$$

6:      **end for**
7:      $\mathcal{H}$-**player:** Call $\mathrm{OPT}_{\mathcal{H}}$ $K$ times on the singleton datasets $\{(x_t, k)\}$ for action $k \in [K]$ with the 0-1 loss, obtaining:

$$h'_k \in \arg\min_{h^* \in \mathcal{H}} \mathbf{1}\{h^*(x_t) \neq k\}$$

8:      $\mathcal{H}$-**player:** Using the $M$ samples $\{(\tilde{g}^{(i)}_t, \tilde{h}^{(i)}_t)\}^M_{i=1}$, construct the $K \times K$ payoff matrix $\tilde{L} \in [-1, 1]^{K \times K}$ indexed by $(k, y) \in [K] \times [K]$:

$$\tilde{L}_{k,y} := \sum_{i=1}^{M} \tilde{\ell}_{x_t}((\tilde{g}^{(i)}_t, \tilde{h}^{(i)}_t), (h'_k(x_t), y)).$$

9:      $\mathcal{H}$-**player:** Solve the linear program

$$\min_{\boldsymbol{p} \in \mathbb{R}^K, \lambda \in \mathbb{R}} \lambda$$
$$\text{s.t. } \boldsymbol{p}^\top \tilde{L} e_y \leq \lambda \quad \forall y \in [K]$$
$$\boldsymbol{p}_k \geq 0 \quad \forall k \in [K]$$
$$\sum \boldsymbol{p}_i = 1$$

     (where $e_y$ is the $y$-th coordinate vector in $\mathbb{R}^K$)
10:      Sample $k \sim \boldsymbol{p}$, let $h_t = h'_k$.
11:      Learner commits to the action $\hat{y}_t = h_t(x_t)$; Nature reveals $y_t$.
12:      Learner incurs the loss $\ell(\hat{y}_t, y_t)$.
13: **end for**

---

**Theorem C.1.** *Let $\mathcal{Y} = \{1, \ldots, K\}$ be a $K$-class action space, $\mathcal{H} \subseteq \mathcal{Y}^{\mathcal{X}}$ be a $K$-valued hypothesis class, $\mathcal{G} \subseteq 2^{\mathcal{X}}$ be a (possibly infinite) collection of groups, and $\ell : \mathcal{Y} \times \mathcal{Y} \to [0, 1]$ be a bounded loss function. Let the Natarajan dimension [Nat89] of $\mathcal{H}$ and the VC dimension of $\mathcal{G}$ both be bounded by $d$. Let $\alpha \geq 0$ be the approximation error of the oracle $\mathrm{OPT}^{\alpha}_{(\mathcal{G}, \mathcal{H})}$. If we are in the $\sigma$-smooth online learning setting, then, for appropriate choices of $M \in \mathbb{N}, n \in \mathbb{N}$, and $\eta > 0$, Algorithm 5 achieves,*

*for each $g \in \mathcal{G}$:*

$$\mathbb{E}[\mathrm{Reg}_T(\mathcal{H}, g)] \leq O\left(\sqrt{\frac{dT \log T}{\sigma}} + \alpha T\right),$$

*where the expectation is over all the randomness of the $(\mathcal{G}, \mathcal{H})$-player's perturbations.*

## C.2    Group-dependent regret

In this section, we give the details on how to achieve the desired *group-dependent regret* guarantees of Section 5. Throughout this section, for any $g \in \mathcal{G}$, let $T_g := \sum_{t=1}^T g(x_t)$. The results in this section hinge on the *GFTPL with small-loss bound* algorithm of [Wan+22]. The main idea will be to instantiate the $(\mathcal{G}, \mathcal{H})$-player in our meta-algorithm, Algorithm 4 using the GFTPL with small-loss bound algorithm so Theorem B.4 allows us to directly inherit its regret guarantee. We quote the algorithm here, adapted to our setting, for reference. Throughout this section, we use the familiar notation from Section 4.2 of the main body:

$$\tilde{\ell}_x((\tilde{g}, \tilde{h}), (y', y)) := \tilde{g}(x)\left(\ell(y', y) - \ell(\tilde{h}(x), y)\right), \tag{20}$$

where $\ell(\cdot, \cdot)$ is the fixed loss of the entire multi-group online learning setting.

---

**Algorithm 6** GFTPL with small-loss bound

---

**Input:** Perturbation matrix $\Gamma \in [-1, 1]^{|\mathcal{G}||\mathcal{H}| \times N}$
 1: Draw i.i.d. vector $\nu = (\nu^{(1)}, \ldots, \nu^{(N)}) \sim \mathrm{Lap}(1)^N$, i.e., $p(\nu^{(i)}) = \frac{1}{2}\exp(-|\nu^{(i)}|)$.
 2: **for** $t = 1, 2, 3, \ldots, T$ **do**
 3:     Set $\nu_t \leftarrow \frac{\nu}{\eta_t}$ where $\eta_t > 0$ is a parameter computed online.
 4:     Using the entire history up to $t - 1$ so far, call $\mathrm{OPT}_{(\mathcal{G}, \mathcal{H})}^\alpha$ to obtain $(\tilde{g}, \tilde{h}) \in \mathcal{G} \times \mathcal{H}$ sastisying:

$$\sum_{s=1}^{t-1} \tilde{\ell}_{x_s}((\tilde{g}, \tilde{h}), (\hat{y}_s, y_s)) + \langle \Gamma^{(\tilde{g}, \tilde{h})}, \nu_t \rangle$$

$$\geq \sup_{(g^*, h^*) \in \mathcal{G} \times \mathcal{H}} \sum_{s=1}^{t-1} \tilde{\ell}_{x_s}((g^*, h^*), (\hat{y}_s, y_s)) + \langle \Gamma^{(\tilde{g}, \tilde{h})}, \nu_t \rangle - \alpha, \tag{21}$$

        where $\Gamma^{(\tilde{g}, \tilde{h})}$ is the $(\tilde{g}, \tilde{h})$th row of $\Gamma$.
 5: **end for**

---

In our setting, the classical FTPL algorithm of [KV05] draws $|\mathcal{G}| \times |\mathcal{H}|$ independent perturbations at each round, which requires enumeration of both $\mathcal{G}$ and $\mathcal{H}$. In order to remedy this, the GFTPL algorithm of [Dud+20] uses a $|\mathcal{G}||\mathcal{H}| \times N$ perturbation matrix $\Gamma$ to generate dependent "shared randomness." The transformation $\Gamma$ applies to a perturbation vector $\nu \in \mathbb{R}^N$, where $N$ is much smaller than $|\mathcal{G}||\mathcal{H}|$. Running FTPL with these perturbations, then, results on an oracle-efficient algorithm. [Wan+22] extends this by showing that, under certain conditions on $\Gamma$, this oracle-efficient algorithm can also achieve a *small-loss regret*, where the regret diminishes based on the total magnitude of the losses over the $T$ rounds instead of the number of rounds.

Specifically, a *small loss regret* looks like the following. It is well-known that, in the worst-case, a regret of $O\left(\sqrt{T \log |\mathcal{G}||\mathcal{H}|}\right)$ is minimax optimal [CL06]. However, stronger bounds have been obtained for problems with "small losses" (see, e.g., [HP05; GSV14; LS15]), where, for a loss function $f : (\mathcal{G} \times \mathcal{H}) \times \mathcal{Y} \to [0, 1]$, one can achieve:

$$O\left(\sqrt{\sum_{t=1}^T f((h_t, g_t), y_t) \log |\mathcal{H}||\mathcal{G}|}\right),$$

which is sharper than the $O\left(\sqrt{T \log |\mathcal{H}||\mathcal{G}|}\right)$ bound when $f((h_t, g_t), y_t) < 1$ on some rounds. We desire this property to achieve a regret bound in the multi-group online learning setting that is sublinear in terms of the number of rounds a group appears $T_g$.

Following [Wan+22], we require the perturbation matrix $\Gamma$ to have two sufficient conditions for Algorithm 6 to obtain the desired small-loss regret. The first is $\gamma$-approximability, which is a condition that ensures the stability choices of $(\tilde{g}_t, \tilde{h}_t)$ and $(\tilde{g}_{t+1}, \tilde{h}_{t+1})$ across rounds. In particular, the stability needed is a bound on the ratio:

$$\frac{\mathbb{P}[(\tilde{g}_t, \tilde{h}_t) = (g, h)]}{\mathbb{P}[(\tilde{g}_{t+1}, \tilde{h}_{t+1}) = (g, h)]} \leq \exp(\gamma \eta_t)$$

for all $(g, h) \in \mathcal{G} \times \mathcal{H}$, where $\eta_t > 0$ is the per-round leraning rate of GFTPL. By Lemma 2 of [Wan+22], the following condition is sufficient to ensure this property. We restate it here, translated to our setting.

**Definition C.1** ($\gamma$-approximability [Wan+22]). *Let $\Gamma \in [-1, 1]^{|\mathcal{G}||\mathcal{H}| \times N}$, where $N$ is the dimension of the noise vector, $\nu$ in Algorithm 6. Define $B_\gamma^1 := \{s \in \mathbb{R}^N : \|s\|_1 \leq \gamma\}$. $\Gamma$ is $\gamma$-approximable if, for all $(g, h) \in \mathcal{G} \times \mathcal{H}$ and $(x, y', y) \in \mathcal{X} \times \mathcal{Y} \times \mathcal{Y}$, there exists $s \in \mathbb{R}^N$ with $\|s\|_1 \leq \gamma$ such that the following holds for all $(g', h') \in \mathcal{G} \times \mathcal{H}$:*

$$\langle \Gamma^{(g,h)} - \Gamma^{(g',h')}, s \rangle \geq \tilde{\ell}_x((g, h), (y', y)) - \tilde{\ell}_x((g', h'), (y', y)).$$

The second property is implementability. This property actually allows us to use our optimization oracle (in our case, $\text{OPT}_{(\mathcal{G}, \mathcal{H})}^\alpha$) to access $\Gamma$ without explicitly representing it. In essence, it requires that we can generate a small number of "fake examples" that effectively implement the perturbation needed by Algorithm 6.

**Definition C.2** (Implementability [Dud+20]). *A matrix $\Gamma \in [-1, 1]^{|\mathcal{G}||\mathcal{H}| \times N}$ is implementable with complexity $M$ if for each $j \in [N]$ there exists a dataset $S_j$ with $|S_j| \leq M$ such that, for all pairs of rows $(g, h), (g', h') \in \mathcal{G} \times \mathcal{H}$,*

$$\Gamma^{((g,h),j)} - \Gamma^{((g',h'),j)} = \sum_{(w,(x,y,y')) \in S_j} w \left( \tilde{\ell}_x((\tilde{g}, \tilde{h}), (y', y)) - \tilde{\ell}_x((\tilde{g}', \tilde{h}'), (y', y)) \right).$$

In the above definition, the fake examples are tuples of a context $x \in \mathcal{X}$ and two outcomes $y, y' \in \mathcal{Y}$. Finally, with the sufficient conditions of implementability and approximability, we quote the main regret guarantee of Algorithm 6in [Wan+22] here.

**Theorem C.2** (Regret guarantee of 6 [Wan+22]). *Let $\{1, \dots, K\}$ be the action space of the Learner and let $\mathcal{Z}$ be the action space of the adversary. Suppose that, at each round, the Learner chooses action $x_t \in [K]$, the adversary chooses action $z_t \in \mathcal{Z}$, and the loss function is $f : [K] \times \mathcal{Z} \to [0, 1]$. Let $L_T^* = \min_{k \in [K]} \sum_{t=1}^T f(k, y_t)$. Then, if there exists a $\gamma$-approximable matrix $\Gamma$, Algorithm 6 instantiated with $\Gamma$ and $\eta_t := \min\left\{ \frac{1}{\gamma}, \frac{C}{\sqrt{L_{t-1}^* + 1}} \right\}$ achieves the following regret bound:*

$$\mathbb{E}[\text{Reg}_T] \leq \left( \frac{4\sqrt{2} \max\{2 \log K, \sqrt{N \log K}\}}{C} + 2\gamma \left( C + \frac{1}{C} \right) \right) \sqrt{L_T^* + 1}$$

$$+ 8\gamma \log \left( \frac{1}{C} \sqrt{L_T^* + 1} + \gamma \right) + 2\gamma^2 + 4\sqrt{2} \max\{2 \log K, \sqrt{N \log K}\} \gamma.$$

*With an appropriate choice of $C > 0$, we may obtain the regret bound:*

$$O\left( \sqrt{L_T^*} \max\left\{ \gamma, \log K, \sqrt{N \log K} \right\} \right) \tag{22}$$

*If $\Gamma$ is also implementable with complexity $M$, then Algorithm 6 is oracle-efficient, making $O(T + NM)$ oracle calls per round, where $N$ is the number of columns of $\Gamma$.*

Finally, we need one more lemma using a standard uniform convergence argument to bound the approximation error from sampling with $\text{OPT}_{(\mathcal{G}, \mathcal{H})}^\alpha$ $M$ times. This is essentially the same as Lemma B.6, but we obtain a sharper bound on $\mathbb{E}\left[ \sum_{t=1}^T \epsilon_t(M) \right]$ at the cost of making polynomially (in $T$) more calls to the oracle.

**Lemma C.3.** *Let $t \in [T]$ and $x_t \in \mathcal{X}$ be fixed, and consider the function $\tilde{\ell}_{x_t}((g, h), (y', y)) := g(x_t)(\ell(y', y) - \ell(h(x_t), y))$. Let $|\mathcal{Y}| = k < \infty$. If $M \geq T^2 \log(k^2 T)$, then over the randomness of drawing $M$ samples $\{(\tilde{g}_t^{(i)}, \tilde{h}_t^{(i)})\}_{i=1}^M$ to construct the empirical distribution $\tilde{q}_t$ described in Definition B.4, for all $y', y \in \mathcal{Y}$, let $\epsilon_t(M)$ be defined as the supremum*

$$\epsilon_t(M) := \sup_{(y', y) \in \mathcal{Y} \times \mathcal{Y}} \left| \mathbb{E}_{(\tilde{g}, \tilde{h}) \sim \tilde{q}_t}[\tilde{\ell}_{x_t}((\tilde{g}, \tilde{h}), (y', y))] - \mathbb{E}_{(g, h) \sim q_t}[\tilde{\ell}_{x_t}((g, h), (y', y))] \right|,$$

*and, over all $T$ rounds,*

$$\mathbb{E}\left[ \sum_{i=1}^T \epsilon_t(M) \right] \leq 2.$$

*Proof.* The proof of this lemma follows the proof of Lemma B.6 exactly, except for the choice of $M$. Therefore, just using the exact same notation as Lemma B.6, we have:

$$\mathbb{P}\left[ \sup_{(y', y) \in \mathcal{Y}^2} \left| \frac{1}{M} \sum_{i=1}^M Z_i(y', y) \right| \geq \varepsilon \right] \leq k^2 \exp(-2M\varepsilon^2)$$

By the same exact argument using $\mathbb{E}[X] \leq \int_0^\infty \mathbb{P}[X \geq t] dt$, we obtain

$$\mathbb{E}\left[ \sup_{(y', y) \in \mathcal{Y}^2} \left| \frac{1}{M} \sum_{i=1}^M Z_i(y', y) \right| \right] \leq w + k^2 \exp(-2Mw^2),$$

where $w > 0$ is an arbitrary parameter. Set $w = \frac{1}{T^2}$. Then, if $M \geq T^2 \log(kT)$, we get

$$\mathbb{E}[\epsilon_t(M)] = \mathbb{E}\left[ \sup_{(y', y) \in \mathcal{Y}^2} \left| \frac{1}{M} \sum_{i=1}^M Z_i(y', y) \right| \right] \leq 2/T.$$

The lemma follows from summing over $T$. $\qquad\square$

*Proof of Theorem 5.1.* We can now prove Theorem 5.1. The main idea is that the small loss regret translates directly into a $o(T_g)$ regret due to how we defined our loss function, $\tilde{\ell}_x$, so we simply instantiate the $(\mathcal{G}, \mathcal{H})$-player in the general algorithm template of Algorithm 4 with Algorithm 6. This allows us to directly inherit the $o(T_g)$ regret guarantee. We restate it here, with parameters specified, as Proposition C.3.1. $\qquad\square$

**Proposition C.3.1** (Theorem 5.1, with parameters specified). *Assume $\mathcal{H}, \mathcal{G}$ are finite and there exists a $\gamma$-approximable and implementable perturbation matrix $\Gamma \in [-1, 1]^{|\mathcal{G}||\mathcal{H}| \times N}$. Let $|\mathcal{Y}| = k$. Let $\alpha \geq 0$ be the approximation parameter of $\mathrm{OPT}_{(\mathcal{G}, \mathcal{H})}^\alpha$. Let the no-regret algorithm for the $(\mathcal{G}, \mathcal{H})$-player in Algorithm 1 be the GFTPL algorithm of [Wan+22] instantiated with $\Gamma$, with parameter $M = T^2 \log(k^2 T)$. Then, for each $g \in \mathcal{G}$:*

$$\mathbb{E}[\mathrm{Reg}_T(\mathcal{H}, g)] \leq O\left( \sqrt{T_g} \max\left\{ \gamma, \log |\mathcal{H}||\mathcal{G}|, \sqrt{N \log |\mathcal{H}||\mathcal{G}|} \right\} + \alpha T \right)$$

*Proof.* Armed with approximability and implementability, we are ready to prove Theorem 5.1. Suppose that there exists a $\gamma$-approximable and implementable perturbation matrix $\Gamma \in [-1, 1]^{|\mathcal{G}||\mathcal{H}| \times N}$. In the setting of Theorem C.2, we instantiate $K = |\mathcal{G}||\mathcal{H}|$, $\mathcal{Z} = \mathcal{X} \times \mathcal{Y} \times \mathcal{Y}$, and the loss function $f(\cdot, \cdot)$ as:

$$f((g, h), (x, y', y)) := \tilde{\ell}_x((g, h), (y', y)) = g(x) \left( \ell(y', y) - \ell(h(x), y) \right).$$

Observe that, with the loss instantiated as $\tilde{\ell}_x$, we have:

$$
\begin{aligned}
L_T^* &= \min_{(g,h)\in\mathcal{G}\times\mathcal{H}} \sum_{t=1}^{T} \tilde{\ell}_x((g,h),(x_t,y_t',y_t)) \\
&\leq \sum_{t=1}^{T} \tilde{\ell}_x((g,h),(x_t,y_t',y_t)) \quad \text{for all } (g,h)\in\mathcal{G}\times\mathcal{H} \\
&= \sum_{t=1}^{T} g(x_t)(\ell(y_t',y_t)-\ell(h(x_t),y_t)) \quad \text{for all } (g,h)\in\mathcal{G}\times\mathcal{H}. \\
&\leq \sum_{t=1}^{T} g(x_t) = T_g,
\end{aligned}
$$

for all $g\in\mathcal{G}$. The last inequality just comes from the fact that $\ell(\cdot,\cdot)\in[0,1]$. By directly applying Theorem C.2, we obtain the regret guarantee for Algorithm 6 of

$$
\begin{aligned}
\mathbb{E}[\text{Reg}_T] &\leq O\left(\sqrt{L_T^*}\max\left\{\gamma,\log|\mathcal{G}||\mathcal{H}|,\sqrt{N\log|\mathcal{G}||\mathcal{H}|}\right\}\right) \\
&\leq O\left(\sqrt{T_g}\max\left\{\gamma,\log|\mathcal{G}||\mathcal{H}|,\sqrt{N\log|\mathcal{G}||\mathcal{H}|}\right\}\right).
\end{aligned}
$$

However, this is just the regret guarantee of Algorithm 6, not the regret guarantee of our end-to-end multi-group online learning algorithm, Algorithm 1. We replace the algorithm of [Blo+22] in Algorithm 1 with Algorithm 6. That is, we use the Algorithm 6 for the $(\mathcal{G},\mathcal{H})$-player in Algorithm 4; a full description of this substitution is in Algorithm 7. By our meta-theorem, Theorem B.4, this entire algorithm achieves the multi-group regret guarantee, for all $g\in\mathcal{G}$:

$$
\mathbb{E}[\text{Reg}_T(\mathcal{H},g)] \leq \sum_{t=1}^{T} v_t^A + \sum_{t=1}^{T} \mathbb{E}[\epsilon_t(M)] + R(T). \tag{23}
$$

$$
\leq \sum_{t=1}^{T} \mathbb{E}[\epsilon_t(M)] + O\left(\sqrt{T_g}\max\left\{\gamma,\log|\mathcal{G}||\mathcal{H}|,\sqrt{N\log|\mathcal{G}||\mathcal{H}|}\right\}\right). \tag{24}
$$

Equation (24) follows from applying Lemma B.5 and using the regret bound for Algorithm 6 established above. It remains to ensure that $\sum_{t=1}^{T} \mathbb{E}[\epsilon_t(M)]\leq o(T_g)$. Simply applying Lemma B.6 results in $\sum_{t=1}^{T} \mathbb{E}[\epsilon_t(M)]=2\sqrt{T}$, which is insufficient for our purposes. Instead, we use Lemma C.3, which ensures that $\sum_{t=1}^{T} \mathbb{E}[\epsilon_t(M)]\leq O(1)$ at the cost of increasing $M$ to be $M\geq T^2\log(k^2T)$, making polynomially more oracle calls to $\text{OPT}_{(\mathcal{G},\mathcal{H})}^{\alpha}$ per-round. This gives us the final regret guarantee of

$$
\mathbb{E}[\text{Reg}_T(\mathcal{H},g)] \leq 2 + O\left(\sqrt{T_g}\max\left\{\gamma,\log|\mathcal{G}||\mathcal{H}|,\sqrt{N\log|\mathcal{G}||\mathcal{H}|}\right\}\right),
$$

as desired. $\square$

One possible setting in which a $\Gamma$ matrix is easily constructible is the *transductive setting.* Here, we explicitly show how to construct $\Gamma$ to obtain Corollary 5.1.1.

*Proof of Corollary 5.1.1.* Let $X\subseteq\mathcal{X}$ be the set the adversary fixes beforehand in the transductive setting, where $N:=|X|$. We can construct a 1-approximable and implementable $\Gamma\in[-1,1]^{|\mathcal{G}||\mathcal{H}|\times 4N}$ by creating a row for each $(g,h)\in\mathcal{G}\times\mathcal{H}$ and a column for each $(x,y,y')\in X\times\mathcal{Y}\times\mathcal{Y}$, and setting each entry as

$$
\Gamma^{((g,h),(x,y,y'))} := \tilde{\ell}_x((g,h),(y',y)).
$$

$\square$

**Algorithm 7** Algorithm for Group Oracle Efficiency (with GFTPL)

---

**Input:** Perturbation matrix $\Gamma \in [-1, 1]^{|\mathcal{G}||\mathcal{H}| \times N}$; number of $\mathrm{OPT}^\alpha_{(\mathcal{G},\mathcal{H})}$ calls $M \in \mathbb{N}$.

1: **for** $t = 1, 2, 3, \ldots, T$ **do**
2:      Receive a (possibly adversarial) context $x_t \sim \mu_t$ from Nature.
3:      **for** $i = 1, 2, 3, \ldots, M$ **do**
4:         $(\mathcal{G}, \mathcal{H})$-**player:** Draw i.i.d. vector $\nu = (\nu^{(1)}, \ldots, \nu^{(N)}) \sim \mathrm{Lap}(1)^N$, i.e., $p(\nu^{(i)}) = \frac{1}{2}\exp(-|\nu^{(i)}|)$.
5:         $(\mathcal{G}, \mathcal{H})$-**player:** Set $\nu_t \leftarrow \frac{\nu}{\eta_t}$ where $\eta_t := \min\left\{\frac{1}{\gamma}, \frac{C}{\sqrt{L^*_{t-1}+1}}\right\}$.
6:         $(\mathcal{G}, \mathcal{H})$-**player:** Using the entire history $\{(\hat{y}_s, y_s)\}_{s=1}^{t-1}$ so far, call $\mathrm{OPT}^\alpha_{(\mathcal{G},\mathcal{H})}$ to obtain $(\tilde{g}, \tilde{h}) \in \mathcal{G} \times \mathcal{H}$ satisfying:

$$\sum_{s=1}^{t-1} \tilde{\ell}_{x_s}((\tilde{g}, \tilde{h}), (\hat{y}_s, y_s)) + \langle \Gamma^{(\tilde{g}, \tilde{h})}, \nu_t \rangle$$

$$\geq \sup_{(g^*, h^*) \in \mathcal{G} \times \mathcal{H}} \sum_{s=1}^{t-1} \tilde{\ell}_{x_s}((g^*, h^*), (\hat{y}_s, y_s)) + \langle \Gamma^{(\tilde{g}, \tilde{h})}, \nu_t \rangle - \alpha, \quad (25)$$

        where $\Gamma^{(\tilde{g}, \tilde{h})}$ is the $(\tilde{g}, \tilde{h})$th row of $\Gamma$.
7:      **end for**
8:      $\mathcal{H}$-**player:** Call $\mathrm{OPT}_{\mathcal{H}}$ twice on the singleton datasets $\{(x_t, -1)\}$ and $\{(x_t, 1)\}$ with the 0-1 loss, obtaining:

$$h'_1 \in \underset{h^* \in \mathcal{H}}{\arg\min} \, \mathbf{1}\{h^*(x_t) \neq 1\} \quad h'_{-1} \in \underset{h^* \in \mathcal{H}}{\arg\min} \, \mathbf{1}\{h^*(x_t) \neq -1\}.$$

9:      $\mathcal{H}$-**player:** Solve the linear program

$$\min_{p, \lambda \in \mathbb{R}} \lambda$$

$$\text{s.t.} \sum_{i=1}^{M} p\tilde{\ell}_{x_t}((\tilde{g}_t^{(i)}, \tilde{h}_t^{(i)}), (h'_1(x_t), y)) + (1-p)\tilde{\ell}_{x_t}((\tilde{g}_t^{(i)}, \tilde{h}_t^{(i)}), (h'_{-1}(x_t), y)) \leq \lambda \quad \forall y \in \{-1, 1\}$$

$$0 \leq p \leq 1.$$

10:     Sample $b \sim \mathrm{Ber}(p)$ where $b \in \{-1, 1\}$, let $h_t = h'_b$.
11:     Learner commits to the action $\hat{y}_t = h_t(x_t)$; Nature reveals $y_t$.
12:     Learner incurs the loss $\ell(\hat{y}_t, y_t)$.
13: **end for**

---

