# OpenReview forum: "Group-wise oracle-efficient algorithms for online multi-group learning"
_NeurIPS.cc/2024/Conference — NeurIPS 2024 poster_

### Official Review · Reviewer_TY8p · 2024-07-07

**Soundness:** 3
**Presentation:** 4
**Contribution:** 4
**Rating:** 7
**Confidence:** 3

**Summary:**

Multi-group learning has been attracted attention and solutions for small-sized groups are already available. This paper further studies the online learning case when the group $\mathcal{G}$ is large or even possibly infinite. In addition to designing an algorithm, the paper also provides extensive theoretical analysis on the results in three settings: the i.i.d. setting, the adversarial setting with smoothed context distributions, and the adversarial transductive setting.

**Strengths:**

1. The paper studies an interesting and important problem. The size of possible groups may indeed be very large in real cases, and infinite as the sample size infinitely grows.

2. The paper, while using relatively advanced mathematics, tackles the problem with well-known and intuitive techniques in optimization and game theory, and consequently achieves satisfying theoretical results.

3. The presentation is very readable: the definitions, results, and remarks are presented in a nice order and achieves the level of rigor for a theoretical paper.

Overall I am in favor of this work because of the elegant solutions for an important problem.

**Weaknesses:**

1. The theoretical analysis is restricted on binary case, i.e. $\mathcal{Y} = \{ -1, 1 \}$. I am aware that the multi-class case would be much difficult to analyze, but this certainly shall be one of the possible directions in the future.

2. (minor) Definition 4.1 uses measure theory that is in a graduate-level course in real analysis and is likely a rarely appearing concept for the majority audience. To keep the paper self-contained, it might be helpful to thoroughly define all concepts such as essential supremum and the differentiation of measures.

**Questions:**

I do not have any questions at this moment.

**Limitations:**

Yes.

---

> ### Author Rebuttal · Authors · 2024-08-06
>
> Thank you for your insight and helpful review. Your comments will be very helpful in improving the presentation of the final paper.
>
> To your first point about the multi-class case, we actually do address this in Appendix C. In the main body, we only included a Remark on line 248, but Appendix C includes all the details for a finite, $K$-valued discrete action space $\mathcal{Y}$. Our main theoretical guarantee for such a case is Theorem C.1, and the algorithm for this case is Algorithm 4.
>
> To your second point about the measure theoretic Definition 4.1, we agree that this may be overly technical. We do need the definition to properly define our setting, but we will take care to define essential supremum and absolute continuity. We will also emphasize that thinking about the results in terms of our simple example where the base measure $\mathcal{B}$ is uniform on $\mathcal{X}$ is sufficient for understanding our results.

---

> > ### Comment · Reviewer_TY8p · 2024-08-13
> > **Thanks for the response**
> >
> > I appreciate the concise response by the authors. I am happy to keep my score.

---

### Official Review · Reviewer_sHAP · 2024-07-12

**Soundness:** 3
**Presentation:** 2
**Contribution:** 2
**Rating:** 6
**Confidence:** 4

**Summary:**

This paper studies the problem of online multi-group learning where the algorithm should achieve sublinear regret with respect to all groups. They propose a group-wise oracle-efficient algorithm that avoids enumerating all groups by accessing the group oracle. An $\tilde{O}(\sqrt{dT/\sigma})$ regret bound is derived for $\sigma$-smoothed online learning setting.

**Strengths:**

1. The paper addresses the problem of avoiding enumeration of groups, which improves the efficiency of the algorithm when the space of group is large.
2. The reduction from online multi-group learning to the AMF framework is novel and interesting.

**Weaknesses:**

1. The discussion of related work for online multi-group learning is not adequate. It would be better if the author included the regret bound and online learning setting in the related work. I think listing a table for comparison among these works is better.
2. The motivation of using the AMF framework and FTPL is still not clear and needs more discussion. Moreover, it would be better if the author could clarify the novelty in the analysis.
3. It would be better if the author could also show the lower bound of this problem. Thus we can see how tight regret is obtained and which part can be improved in future work.
4. The computational efficiency of the $(G, H)$-optimization oracle is not discussed. It seems that this oracle is much harder than the ERM problem in [Ach23+] since this oracle optimizes over both $G$ and $H$.

**Questions:**

Please see the weaknesses part.

**Limitations:**

The authors have addressed the limitations and societal impact.

---

> ### Author Rebuttal · Authors · 2024-08-06
>
> Thank you for your insight and helpful review. Your comments will be very helpful in improving the presentation of the final paper. In particular, we appreciate your feedback on the presentation of our results and related work, and we believe that addressing these issues will greatly improve clarity.
>
> To your first point on related work, space limitations did not allow us to include a table in the preliminary submission, but we agree that a table of existing results would help presentation. Please see the attached rebuttal PDF for our draft of this table. For multi-group learning in the online, sequential setting, the only two works that provide comparable regret bounds are, to our knowledge, [BL20] and [Ach+23], which both obtain a $\sqrt{T_g \log |\mathcal{H}| |\mathcal{G}|}$ regret requiring enumeration of $\mathcal{H}$ and $\mathcal{G}$ or $\mathcal{G}$ itself with an oracle for $\mathcal{H}$. We will state this in our table in the camera-ready version, making specific note how the algorithms fare in terms of space and computationally complexity, our main concern. In the online learning setting without group considerations, we can include the existing results for smoothed online learning from [Blo+22] in our table, which, to our knowledge, are the best possible regret bounds for the smoothed setting.
>
> To your second point on AMF and FTPL, we appreciate the feedback and believe it is important that the intuition of these two main ingredients is clear. In a nutshell, the AMF framework allows us to have low-regret with respect to all $\mathcal{G} \times \mathcal{H}$ pairs; the multi-objective regret guarantees are useful for our purposes because we want to guarantee simultaneous low regret over all $\mathcal{G}$. However, naively applying AMF requires us to enumerate these objectives, in turn enumerating $\mathcal{G}$, the main issue we want to avoid. FTPL comes to the rescue and allows us to have low-regret to all $\mathcal{G} \times \mathcal{H}$ pairs implicitly without enumerating them. So AMF allows us to compete with all $g \in \mathcal{G}$; FTPL ensures this is efficient. We see that perhaps this needed more description in the Introduction, namely around line 82 in Section 1.1 (Summary of Results), so, in the camera-ready version, we will include the main novelty of using these techniques in the intro. We will also refer the interested reader to "Technical Details" in Section 4.2 for the main high-level description of the analysis and techniques. Then, in the "Technical Details" in Section 4, on line 258 and below, we will make sure this high-level description of our analysis is clear and that each component, the AMF and the FTPL, have a clear purpose.
>
> To your third point about lower bounds, we agree that it is important to state the existing known limits, and we will provide details about existing lower bounds in the camera-ready work where we have more space. Statistical and computational lower bounds exist from [Blo+22], [HK16], and [HHSY22] for the oracle-efficient online learning setting without groups, so they immediately apply to our setting, as the group setting is a strictly harder generalization. We will include such lower bounds in our work. The lower bound of [HK16] motivates the line of work in oracle-efficient online learning, demonstrating that, in the fully adversarial online setting, even when given an oracle for $\mathcal{H}$, regret must be $\Omega(T)$ unless we our time complexity is $\mathrm{poly}(|\mathcal{H}|)$. This motivates working with an optimization oracle in more restricted settings, such as the *smoothed* setting we extensively study in our paper. Lower bounds for the oracle-efficient *smoothed* setting from [Blo+22] without groups show that, in the classification setting we consider, the $\sqrt{\frac{dT \log T}{\sigma}}$ regret is essentially tight. In particular, they show that *any* online algorithm with access to an ERM oracle cannot achieve sublinear regret against a $\sigma$-smooth adversary unless $T \geq \tilde{\Omega}(\sigma^{-1/2})$. So in the smoothed classification setting, our main result is essentially optimal.
>
> To your fourth point, we definitely agree that a main open question in our work is the nature of the $(\mathcal{G}, \mathcal{H})$-optimization oracle. To our knowledge, [GKR22] is the only work to have explicitly stated instantiations for such an oracle. There are many computational intractability results for agnostic learning, and multi-group learning is at least as hard as agnostic learning, so we don't expect to get end-to-end computationally efficient multi-group learning algorithms in general.
>
> To address Reviewer wskr, as well, we believe it would be very helpful to the presentation of the paper to include the algorithm description for the $(\mathcal{G}, \mathcal{H})$-optimization oracle in our paper so it can be easily referenced. Currently, we only make a passing mention to [GKR22], but we will include a lengthier discussion in the camera-ready version. There are two existing algorithms that reduce optimizing over $(\mathcal{G}, \mathcal{H})$ to weighted multi-class classification or alternating ERM over $\mathcal{G}$ and $\mathcal{H}$. Both suggest implementable heuristics in practice. In the camera-ready version, we will include the algorithms for the ternary classification oracle and the alternating minimization oracle, as well as their main guarantees from [GKR22] in the appendix or full body. We will also briefly discuss some heuristic possibilities for instantiating such oracles. We note that, due to the computational hardness of ERM in the worst-case, the oracle in [Ach+23] requires access to similar heuristics.
>
> For more on the computational efficiency of this oracle, please see our response to Reviewer wskr. We believe that we have addressed all the weaknesses in this response and will make the appropriate additions in the camera-ready version. Please let us know if anything was insufficient or unclear.

---

> > ### Comment · Reviewer_sHAP · 2024-08-12
> >
> > Thanks for your response. You have addressed my concerns about the related work, lower bound, and computational efficiency of the oracle. Hence, I will raise my score to 6.

---

### Official Review · Reviewer_wskr · 2024-07-12

**Soundness:** 4
**Presentation:** 4
**Contribution:** 3
**Rating:** 7
**Confidence:** 5

**Summary:**

The paper provides oracle efficient algorithms for online multi-group learning. The goal is to obtain sublinear regret for each group subsequence $g \in G$ simultaneously.
[BL20] showed that $o(T_g)$ regret is possible with finite $H$ and $G$ but must enumerate both $H$ and $G$. [Ach+23] showed $o(T_g)$ regret while being oracle efficient in the hypothesis class, but must enumerate over $G$.  In this work, the authors use an optimization oracle for $G \times H$ jointly and avoid explicit enumeration for either G or H.

To do so, they reduce the multigroup online learning to Adversary moves first framework of [Lee+22], and adapt results from oracle efficient online learning in the standard (no-groups) setting.

**Strengths:**

- The paper is technically solid and a substantial contribution to online learning with multi group regret gurantees, and is the first to be oracle efficient in both G and H.
- The reduction of multi group regret to AMF regret and the resulting minmax game simplification is novel.
- They provide multigroup regret analogs of smoothed online learning, generalized FTPL.

**Weaknesses:**

- For the $G \times H$ oracle mentioned in definition 2.2 the authors defer to [GKR22] who in turn provide two constructions: one based on ternary classification and another alternating minimization heuristic.  It would be beneficial to add a longer discussion on this in the main body or the appendix to make the paper self contained.
- Computational efficiency of the $G \times H$ oracle, this hasn't been discussed explicitly, are there group structures for which the optimization over G is efficient and doesn't involve enumeration?Algorithm 6 in [GKR22], doesn't discuss this either, it seems like without any further assumptions, enumeration is the best we can do in practice.

**Questions:**

Please address the above.

Also, the appendix describes multi-class action space, what would change with real valued labels?

**Limitations:**

The authors discuss limitations of their paper.

---

> ### Author Rebuttal · Authors · 2024-08-06
>
> Thank you for your insight and helpful review. Your comments will be very helpful in improving the presentation of the final paper.
>
> We definitely agree that a main open question in our work is the nature of the $(\mathcal{G}, \mathcal{H})$-optimization oracle. To our knowledge, [GKR22] is the only work to have explicitly stated instantiations for such an oracle. On a high-level note, our work follows a long line of work that exploit oracle primitives *assumed* to be computationally efficient (referenced in our work), so we don't expect to get end-to-end computational efficiency. There are a plethora of well-known hardness results for agnostic learning with various concept classes, and multi-group learning is a generalization of agnostic learning, so the need to assume such primitives is necessary in many cases (and models the empirical success of ERM despite hardness results).
>
> To your first point in "Weaknesses," we agree that it would be very helpful to the presentation of the paper to include the algorithm description for the $(\mathcal{G}, \mathcal{H})$-optimization oracle in our paper so it can be easily referenced. In the camera-ready version, we will include the algorithms for the ternary classification oracle and the alternating minimization oracle, as well as their main guarantees from [GKR22] in the appendix or main body. We will also briefly discuss some heuristic possibilities for instantiating such an oracle.
>
> To your second point in "Weaknesses," it is not entirely clear to us what the computational efficiency of such an oracle would be, at least in the theoretical sense. From a heuristics standpoint, the aforementioned ternary classification instantiation from [GKR22] can be instantiated so long as we have access to weighted multi-class classification, which, at least in practice, can be implemented with cost-sensitive classification packages. In this case, $\mathcal{G}$ is a byproduct of the class upon which this ternary classifier outputs a "defer" label. On the other hand, if there's a specific $\mathcal{G}$ we want to address, the aforementioned alternating maximization oracle from [GKR22] can be implemented with ERM access to $\mathcal{G}$. Of course, ERM is computationally hard in the worst-case, but heuristics for ERM exist and it is a common assumption in the literature to assume that, at the very least, approximate ERM is feasible. Because $\mathcal{G}$ is class of group indicators, this is equivalent to assuming that we have ERM heuristics for binary classification. However, we should mention that the alternating maximization heuristic can only be guaranteed to supply saddle points, not true global maxima. In both these cases, heuristics (either for multi-class weighted classification or ERM) exist that avoid enumeration. However, it's clear that both these methods have limitations, and we will point out these limitations in the camera-ready version, along with the algorithm descriptions, as we mentioned in the point above. We believe that it is an interesting and worthwhile open question to develop more specific instantiations of this $(\mathcal{G}, \mathcal{H})$-oracle for more specific problem settings that are computationally efficient and have provable optimization guarantees.
>
> To your final point about real-valued action spaces, the main challenge is that our $\mathcal{H}$-player must solve a $|\mathcal{Y}| \times |\mathcal{Y}|$ size linear program, ultimately playing a distribution over possible actions which are of the order $|\mathcal{Y}|$. Because the $\mathcal{H}$-player must hedge against all possible $\mathcal{Y}$ the adversary could select in its linear program, we were only able to implement this step with discrete, finite label spaces. We believe it is an interesting open problem to extend our techniques to real-valued action spaces. One naive route might be to take a discretization approach, but a fine-enough discretization would defeat the purpose of the simple LP the $\mathcal{H}$-player must solve at every round. [Blo+22] does give an approach for the (single group) case for real-valued labels in the smooth setting, but the challenge remains to design a feasible game for the $\mathcal{H}$-player to solve at every round to hedge against the maximally adversarial choice in a continuous $\mathcal{Y}$.

---

> > ### Comment · Reviewer_wskr · 2024-08-11
> >
> > Thanks for the addressing my comments, I stand with my original assessment for accepting this paper.  I appreciate the discussion on oracles for GxH and highlighting this in the main body and providing some concrete though not always computationally efficient heuristic in the appendix would be great, these should also address the other reviewer's concern.

---

### Official Review · Reviewer_K3De · 2024-07-18

**Soundness:** 3
**Presentation:** 2
**Contribution:** 3
**Rating:** 6
**Confidence:** 3

**Summary:**

This paper studies the online multi-group learning problem, and considers it in both smoothed context and general ($\gamma$-approximatable) settings. Unlike the traditional online learning setting, this paper considers achieve low regrets for all groups (subsequences). It utilizes the adversary moves first (AMF) framework for multi-objective online learning from [Lee+22] and the follow-the-perturbed-leader (FTPL) algorithm to handle the large number of groups. Through subtle algorithmic design and analysis, the paper achieved computationally-efficient and oracle-efficient learning.

**Strengths:**

The paper studied the multi-group online learning problem, which is related to fair learning and subgroup robustness literature, both are significant. Under the smoothed setting, it achieves $\tilde{O}(\sqrt{dT/\sigma})$ regret; while under the general setting where $\gamma$-approximability is guaranteed, it achieves $\tilde{O}(\sqrt{NT_{g}\log\lvert H\rvert \lvert G\rvert})$ for each group $g$.

It first considers the i.i.d setting (both offline and online) then quickly moves forward to the smoothed online setting. Under the AMF framework, it considers two players: $(G,H)$-player (adversary that has access to an optimization oracle $\text{OPT}^{\alpha}_{(G,H)}$ that maximizes the learner's loss) and the $H$-player. Under such a game-theoretic perspective, the paper proposes algorithms that achieves bounded regret. It then adapts the algorithm to broader settings (non-smoothed settings), where the $\gamma$-approximability is assumed. For groups with different densities, the algorithm can achieve low regret for each group $g$. The algorithmic designs are subtle and the analysis is sound to the review.

**Weaknesses:**

The main text in Section 2 and after are fine, while Section 1 needs improvement. Specifically, some descriptions are unclear.

For example, the bound $\tilde{O}(\sqrt{NT_{g}\log\lvert H\rvert \lvert G\rvert})$ is achieved for each group $g$, but this is not brought forward in summary of result (Line 76) and the last time $g$ is mentioned as one group is in Line 18.

Line 54-55, if $G$ is rich and expressive, then all subgroups well-approximated by a group $g$ in $G$ will be "covered" by whom? What does this sentence mean?

**Questions:**

See Weaknesses.

**Limitations:**

No concerns.

---

> ### Author Rebuttal · Authors · 2024-08-06
>
> Thank you for your insight and helpful review. Your comments will be very helpful in improving the presentation of the final paper.
>
> We will take care in the camera-ready version to make revisions to Section 1 that improve its clarity. In particular, the additional space of a page should allow us to address these issues adequately.
>
> - Lines 54-55: In this line, we mean that if $\mathcal{G}$ is expressive enough, then any potentially relevant subgroup should be well-approximated by some $g \in \mathcal{G}$, and, therefore, any individual $x \in g$ should be ensured that the quality of their predictions are as good as the best possible prediction for their group. When we wrote that those subgroups would be "covered," we meant that those subgroups should be included in, or, at least, well-approximated by some $g \in \mathcal{G}$, so our guarantees apply to that subgroup. We will make this wording more clear in the revision, replacing this clause with "...but multi-group learning with a very rich and expressive family of groups $\mathcal{G}$ ensures that, as long as a subgroup is well-approximated or within the collection $\mathcal{G}$, it obtains our theoretical guarantees. This motivates dealing with large or potentially infinite $\mathcal{G}$ to provide guarantees for as many subgroups as possible."
> - Line 18 + Line 76: Thank you for bringing this to our attention. We will clarify early in the introduction that $g$ (lowercase) denotes a single subset of $\mathcal{X}$. In the "Summary of Results," we will clarify that our regret guarantee is specific to each $g \in \mathcal{G}$ by starting the claim in Line 75 with, "...for each particular group $g \in \mathcal{G}$."
>
> In addition to the comments you raised, we will clarify the descriptions in the Summary of Results using the added space. In particular, we will add sentences describing the parameters in the regret bounds, and, potentially, if there's still space, add a table to compare to results from previous work. We included a draft of such a table in the global Author Rebuttal PDF.
>
> Please let us know, if, in addition to the proposed modifications above, there are any other clarity issues you noticed.

---

> > ### Comment · Reviewer_K3De · 2024-08-11
> > **Official Comment by Reviewer K3De**
> >
> > Thank you very much for the response. Please include the discussions in revision for better readability. I will keep the current review.

---

### Author Rebuttal · Authors · 2024-08-06

We have included a draft of the table that will appear in the Introduction section of the camera-ready version of our work to address Reviewers K3De and sHAP. This table includes a summary of existing results in comparison to ours.

---

### Decision · Program_Chairs · 2024-09-25

**Decision:**

Accept (poster)

**Comment:**

The reviewers are in agreement that this paper is a solid piece of work that designs computationally efficient algorithms for online multi-group learning. In comparison with previous works, this paper avoids the enumeration of groups (and assuming access to a (G,H) offline learning oracle) which results in a reduced computational complexity.

In the final version, the authors are encouraged to incorporate discussions, e.g. comparison table, the feasibility of (G,H) offline learning oracle, and regret lower bounds.